# DSBench: How Far Are Data Science Agents from Becoming Data Science Experts?

**Liqiang Jing**[1,2] *    **Zhehui Huang**[2,3]    **Xiaoyang Wang**[2]    **Wenlin Yao**[2]    **Wenhao Yu**[2]

**Kaixin Ma**[2]    **Hongming Zhang**[2]    **Xinya Du**[1]    **Dong Yu**[2]

[1]University of Texas at Dallas    [2]Tencent AI Lab, Seattle    [3]University of Southern California

## Abstract

Large Language Models (LLMs) and Large Vision-Language Models (LVLMs) have demonstrated impressive language/vision reasoning abilities, igniting the recent trend of building agents for targeted applications such as shopping assistants or AI software engineers. Recently, many data science benchmarks have been proposed to investigate their performance in the data science domain. However, existing data science benchmarks still fall short when compared to real-world data science applications due to their simplified settings. To bridge this gap, we introduce DSBench, a comprehensive benchmark designed to evaluate data science agents with realistic tasks. This benchmark includes 466 data analysis tasks and 74 data modeling tasks, sourced from ModelOff and Kaggle competitions. DSBench offers a realistic setting by encompassing long contexts, multimodal task backgrounds, reasoning with large data files and multi-table structures, and performing end-to-end data modeling tasks. Our evaluation of state-of-the-art LLMs, LVLMs, and agents shows that they struggle with most tasks, with the best agent solving only 34.12% of data analysis tasks and achieving a 34.74% Relative Performance Gap (RPG). These findings underscore the need for further advancements in developing more practical, intelligent, and autonomous data science agents.

## 1 Introduction

Large Language Models (LLMs) (OpenAI, 2023a; Touvron et al., 2023b) and Large Vision-Language Models (LVLMs) (OpenAI, 2023b; Liu et al., 2023b) have achieved compelling success on various vision and language tasks, such as natural language understanding (Wang et al., 2019), visual question answering (Antol et al., 2015), and image captioning(Lin et al., 2014), demonstrating their adaptability and effectiveness. However, despite their achievements, LLMs and LVLMs face limitations when applied to certain real-world tasks due to the lack of integration with practical applications, such as computer manipulation. To address this, advanced LLMs and LVLMs are increasingly being incorporated into interactive intelligent systems, enabling them to tackle complex tasks with additional tools and interfaces. A prominent example of this is the data science agent, an emerging research area that assists individuals and organizations in making informed decisions, predicting trends, and improving processes by analyzing large volumes of data (Hong et al., 2024; Guo et al., 2024; chapyer team, 2023; Zhang et al., 2024c).

The data science agent aims to address data-centric scientific problems, including machine learning, data analysis, and mathematical problem-solving, which present unique challenges, such as complex and lengthy task-handling steps. For example, Jupyter AI (jupyter-ai team, 2023) connects generative language models with Jupyter notebooks[1] and provides a user-friendly and powerful way to improve developer productivity in the Jupyter Notebook. MLCopilot (Zhang et al., 2024a) leverages LLMs to generate solutions for novel real-world machine learning tasks, based on the existing experiences from historical tasks. To evaluate the performance of the data science agent, the existing work focuses on developing either code generation benchmarks (Zhang et al., 2024c; Zan et al., 2022; Chandel et al., 2022) or math problem benchmarks (Lu et al., 2023; Cobbe et al., 2021).

---

*This work is done during Liqiang Jing and Zhehui Huang's internship at Tencent AI Lab, Bellevue, USA.
[1]https://jupyter.org/.

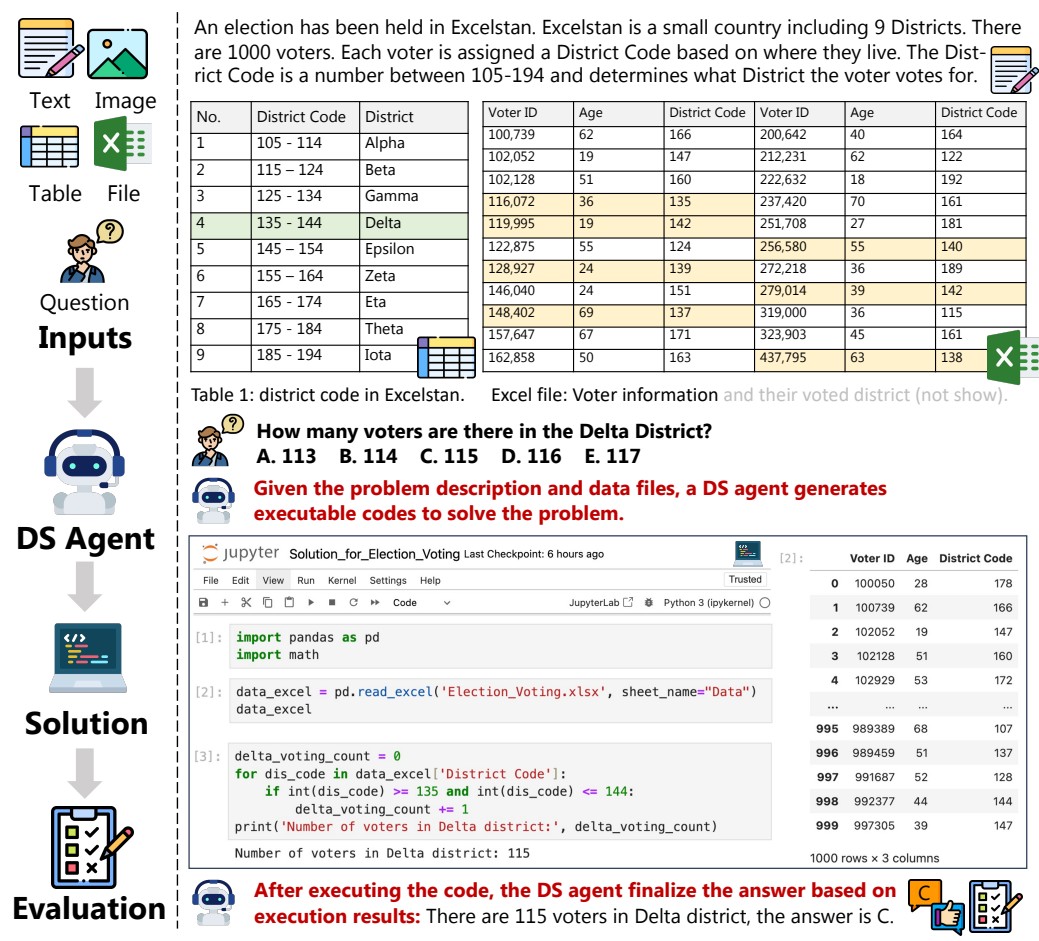

Figure 1: The illustration of the complete workflow of our proposed DSBench benchmark, from task description and data file processing to model or agent execution, followed by the final evaluation. It demonstrates how a data science agent approaches a problem and processes the data, while the evaluation process compares its solution with the ground-truth answer.

For example, DS-1000 (Lai et al., 2023) introduce one thousand code completion problems covering seven widely-used Python data science libraries: NumPy, Pandas, TensorFlow, PyTorch, SciPy, Scikit-learn, and Matplotlib. Hendrycks et al. (2021b) introduce the MATH benchmark, which enables the community to measure the mathematical problem-solving ability of models. Although these benchmarks can be applied to investigate the performance of data science models, they still do not closely reflect real-world data science tasks.

Building an effective benchmark for data science agents/models still presents a significant challenge. The tasks included must be sufficiently complex to simulate real scenarios, yet the predictions made by these models must remain straightforward to verify. Although existing benchmarks achieved compelling success, they are still under a simplified setting compared to real-world data science tasks. Firstly, the task instructions in existing benchmarks are often brief and limited to single modalities. In contrast, real-world tasks typically involve lengthy instructions and multiple modalities. Furthermore, some existing benchmarks only provide incomplete evaluations, focusing primarily on the simple code completion or in-filling capabilities of LLMs/LVLMs, which can be resolved by a few-step reasoning or a few lines of code, overlooking the end-to-end evaluation of the whole system's performance. Additionally, the evaluation of existing benchmarks may be biased because it is limited to certain environments, such as specific Python packages. However, real-life data science tasks are tool-irrelevant and data-centric.

To tackle these limitations, we introduce a comprehensive data science benchmark, DSBench, and the workflow of our benchmark is shown in Figure 1. Our benchmark contains two categories of tasks: *data analysis* and *data modeling*. The former focuses on answering a financial data analysis

Table 1: Comparison with existing agent benchmarks. Columns include the research field (Field), whether the task instruction includes the data file (Data File?), table (Table?), and image (Image?), the word length of the task description (#Len), whether an executable evaluation function is provided (Exec. Eval.?), whether the benchmark asked the agent to finish the task in a fixed/constrained environment (Env. Fix.?), whether the benchmark only asked the agent to generate code (Code-only?) and the number of tasks (Tasks).

| Benchmark | Field | Data File? | Table? | Image? | # Len. | Exec. Eval.? | Code only? | Env. Fix.? | Tasks |
|---|---|---|---|---|---|---|---|---|---|
| Spider (Yu et al., 2018) | Text-to-SQL | ✗ | ✗ | ✗ | - | ✗ | ✗ | ✓ | 1,034 |
| MLAgentBench (Huang et al., 2023) | Machine Learning | ✓ | ✗ | ✗ | - | ✓ | ✗ | ✓ | 13 |
| SWE-Bench (Jimenez et al., 2023) | Software | ✗ | ✗ | ✗ | 195.1 | ✓ | ✗ | ✗ | 2,294 |
| Mind2Web (Deng et al., 2023) | Web | ✗ | ✗ | ✗ | - | ✗ | ✗ | ✗ | 2,000 |
| WEBLINX (Lu et al., 2024) | Web | ✗ | ✗ | ✗ | - | ✗ | ✗ | ✗ | 2337 |
| WorkArena (Drouin et al., 2024) | Web | ✓ | ✓ | ✓ | - | ✓ | ✗ | ✓ | 29 |
| AndroidWorld (Rawles et al., 2024) | Android | ✓ | ✗ | ✓ | - | ✓ | ✗ | ✓ | 116 |
| WebArena (Zhou et al., 2023) | Web | ✗ | ✗ | ✗ | - | ✓ | ✗ | ✓ | 812 |
| OSWorld (Xie et al., 2024) | Computer Control | ✗ | ✗ | ✗ | - | ✓ | ✗ | ✓ | 369 |
| DS-1000 (Lai et al., 2023) | Data Science | ✗ | ✗ | ✗ | 140.0 | ✗ | ✓ | ✗ | 1,000 |
| Arcade (Yin et al., 2023) | Data Science | ✗ | ✗ | ✗ | 18.4 | ✗ | ✓ | ✗ | 10,082 |
| Spider2-V (Cao et al., 2024) | Data Science | ✗ | ✗ | ✗ | - | ✓ | ✗ | ✓ | 494 |
| DSEval (Zhang et al., 2024c) | Data Science | ✓ | ✗ | ✗ | - | ✓ | ✓ | ✗ | 825 |
| DSBench (Ours) | Data Science | ✓ | ✓ | ✓ | 797.9 | ✓ | ✗ | ✗ | 540 |

question that needs the agent to fully understand the data and the question's intent, and questions are either multiple-choice or fill-in-the-blank. The latter requires the agent to build predictive models to learn from the training data and make predictions for the testing data. Specifically, our dataset is built on data competitions ModelOff[2] and Kaggle[3]. In total, we collected 466 data analysis tasks from Modeloff and 74 data modeling tasks from Kaggle. DSBench offers several advantages over existing data science benchmarks. The rationale for using these two platforms is that ModelOff and Kaggle are among the most popular data science competitions, offering tasks that closely resemble real-world scenarios, where each task requires extensive data manipulation. These include a more realistic data science benchmark setting that encompasses understanding long contexts and multi-modal task backgrounds, reasoning with large data files and multi-table structures, and performing end-to-end data modeling, as shown in Table 1.

For data analysis tasks, we mainly utilize the accuracy rate as the metric. On the other hand, for the data modeling tasks, it is non-trivial to investigate their overall performance because of inconsistency in numerical ranges and evaluation dimensions for metrics of different data modeling tasks. Therefore, we further propose the Relative Performance Gap (RPG) to normalize the different metrics in our data modeling tasks. We evaluate multiple state-of-the-art LLMs, LVLMs, and agents, discovering that they fail to solve most of the tasks. The best-performing agent in our experiments achieves only 34.12% accuracy for data analysis tasks and 34.74% RPG for data modeling tasks.

Our contribution can be summarized as follows: (1) We construct a data science benchmark, DS-Bench, which consists of 466 data analysis tasks and 74 data modeling tasks; (2) To comprehensively evaluate existing approaches for the data modeling tasks, we propose the Relative Performance Gap metric that can normalize various evaluation metrics for data modeling; (3) We evaluate representative state-of-the-art LLMs, LVLMs, and agents including the most recent GPT-4o, Claude, and Gemini models, and find that our benchmark is challenging for most of the existing approaches. [4]

## 2 DATA SCIENCE AGENT BENCHMARK

Data science often requires handling complex data, extracting insights, and building models to solve problems. To ensure DSBench reflects these practical demands, we focus on these two task types: data analysis, and data modeling. In our search for appropriate datasets and challenges, we identified

---

[2] https://corporatefinanceinstitute.com/resources/financial-modeling/modeloff-guide/.

[3] https://www.kaggle.com/.

[4] We released all our data and code on Github https://github.com/LiqiangJing/DSBench.

Table 2: Summary of dataset characteristics for data analysis tasks. Len_Intro and Len_Que are the length of the task introduction and questions

| | Mean | Max | Min | Total |
|---|---|---|---|---|
| #Challenges | - | - | - | 38 |
| #Questions | 12.3 | 50 | 3 | 466 |
| Len_Intro | 749.58 | 28,487 | 0 | 28,484 |
| Len_Que | 65.9 | 406 | 6 | 30,691 |
| #Excel | 0.8 | 2 | 0 | 31 |
| Excel Size (KB) | 236.6 | 2,755.9 | 0.2 | 7,333.4 |
| #Image | 0.1 | 1 | 0 | 5 |
| #Sheets | 2.3 | 4 | 1 | 69 |
| #Table | 1.3 | 12 | 0 | 49 |

Table 3: Summary of dataset characteristics on data modeling tasks. File size is the size of Training set.

| | Mean | Max | Min | Total |
|---|---|---|---|---|
| #Competitions | - | - | - | 74 |
| #Metrics | - | - | - | 18 |
| Context Length | 688 | 2,505 | 216 | 50,875 |
| #Training samples | 287k | 4,828k | 200 | 21,270k |
| File Size | 61 GB | 487 GB | 11 KB | 4,519 GB |

that ModelOff and Kaggle provide diverse and realistic tasks that align well with our requirements for data science and data modeling tasks, respectively.

## 2.1 DATA ANALYSIS TASKS

### 2.1.1 DATA COLLECTION

Modeloff is a global financial data analysis competition that challenges contestants to use Excel to solve data-centric questions and case studies. It mostly focuses on independent questions that involve mini exercises in Excel. The questions in Modeloff challenges consist of data analysis tasks that different tools, such as Python, Excel, and Matlab can solve. Therefore, we resort to the Modeloff challenge for the evaluation of data analysis ability in data science agents. We collect all Modeloff challenges and then filter all the challenges that do not contain any questions. Finally, the original 43 challenges are filtered down to 38 challenges with 466 questions. The question types can be categorized into multi-choice questions and fill-in-the-blank questions. The data statics of our data analysis tasks are detailed in Table 2.

### 2.1.2 TASK FORMULATION

**Input and output.** Suppose we have the task introduction $I$, the data files $D = \{d_1, \cdots, d_{N_d}\}$, and the question $Q$, we then feed them into a data science agent $\mathcal{G}$ to answer the question $Q$, *i.e.,* $\hat{A} = \mathcal{G}(I, D, Q)$. $N_d$ is the total number of data files and it can be 1. $\hat{A}$ is the generated answer by the $\mathcal{G}$.

**Evaluation metrics.** To evaluate the performance of the whole agent system, we compare the semantics of ground-truth answer $A$ and the predicted answer $\hat{A}$ by $S(A, \hat{A})$. If the semantics of ground-truth answer $A$ and the predicted answer $\hat{A}$ are the same, we consider the generated answer to have successfully answered the question. $S(\cdot)$ is the semantics comparison function that is implemented by a LLM and prompt in Appendix C. The metric for our benchmark is the percentage of data science questions that are answered correctly, *i.e.,* task-level accuracy. In addition, we also introduce competition-level accuracy for comprehensive evaluation. Competition-level accuracy is calculated by averaging the accuracy scores obtained from each competition.

### 2.1.3 FEATURES

**Various Modalities.** Different from the previous works which mainly focus on textual modality (*e.g.,* (Lai et al., 2023; Cobbe et al., 2021)), our task consists of various modalities, such as images, Excel files, and tables. To show the distribution of the different modalities in different competitions, we visualized the number of competitions in different modalities, as shown in Figure 2(a).

**Complex Table.** Various tables are contained in this dataset. There may be several tables for one question. Therefore, the data science agent must identify which table is important for the current question. Furthermore, some questions require analyzing data across several tables. Unlike previous benchmarks, the tables in this dataset are longer, making it difficult to solve these questions without additional tools, such as Python or Excel, even for humans. In addition, the formats of different

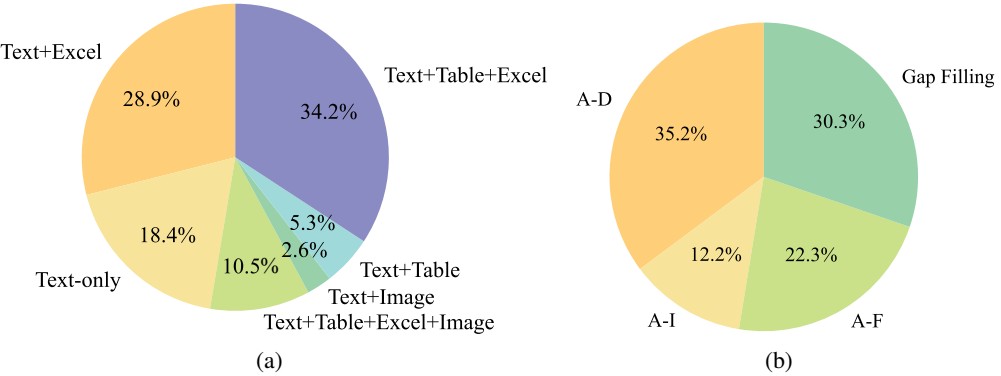

Figure 2: Visualization of the distribution of the datasets. (a) Distribution of data analysis challenges in terms of modalities. Every challenge may contain several questions. (b) Distribution of data analysis challenges in terms of question types.

tables in one challenge are changeable. For example, some tables have properties in their first row, and some tables have properties in their first column.

**Diverse Long Context.** The context in our dataset consists of long textual descriptions (815.44 words on average) as well as multiple modality content. Resolving this kind of data analysis question requires 1) a full understanding of the long description text and the corresponding tables, data files, and images, and 2) the ability to identify the important context semantic content relevant to the current question.

**Wide Scope for Possible Solutions.** Our evaluation task provides a critical platform for assessing the capabilities of data science agents and the corresponding interactive environment. Our data analysis tasks can be utilized to compare a variety of approaches, from pure LLMs/LVLMs to cutting-edge data science agents. The task setting greatly expands the freedom of tool use and encourages developers to employ creative strategies that may diverge from established norms (such as using Excel). For example, we can use either Excel or Python to calculate the amount of tax a company pays.

## 2.2 DATA MODELING TASKS

### 2.2.1 DATA COLLECTION

To evaluate the performance of data science agents on data modeling tasks, we resort to machine learning competitions. Kaggle is a data science competition platform and online community for data scientists and machine learning practitioners. From the platform, we find there are total 648 competitions. Since the testing set in the Kaggle competition is inaccessible, we split the original training set into the training set and testing set as an 8:2 ratio for evaluation. In this way, we could directly get the performance of the solution devised by a data science agent, avoiding submitting the solution to the Kaggle website. To split the dataset easily, we only retain the competitions with a training file, a testing file, and a sample of submission file. Then we can use an automatic code to split the original data files in the Kaggle competition. Finally, we get 74 data modeling competitions in our data science benchmark. All the statics information of our data modeling tasks is detailed in Table 3.

### 2.2.2 TASK FORMULATION

**Input and output.** Suppose we have the competition description $E$, the training set $A$, the testing set $S$, and the sample of the submission file $M$, we then feed them into a data science agent $\mathcal{G}$, which could devise an algorithm and implement the corresponding code to generate the submission file $\hat{F}$ which is the predicted result for the input testing set, *i.e.*, $\hat{F} = \mathcal{G}(E, A, S, M)$. $\hat{F}$ is the generated submission file by the $\mathcal{G}$ and it has a similar format to the sample of the submission file $M$.

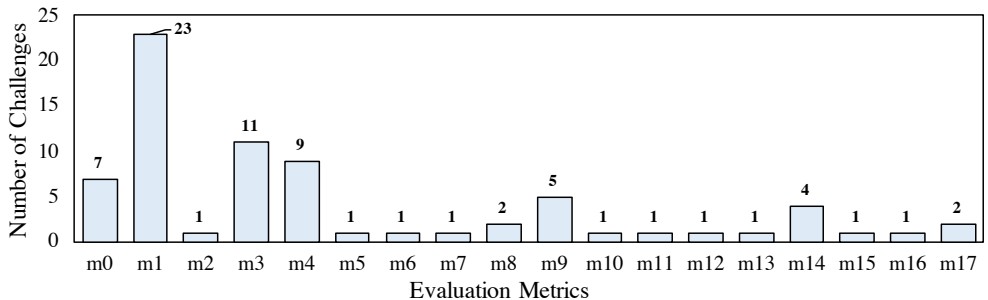

Figure 3: The chart displays the count of Kaggle competitions (vertical axis) categorized by the evaluation metrics used (horizontal axis). Each bar represents the number of competitions that employ a specific metric, highlighting the diversity of evaluation criteria in DSBench. The denotation of "m0-m17" can be found in Appendix D. The top-3 used metrics are ROC ("m1") Root Mean Squared Logarithmic Error ("m3") and Root Mean Squared Error ("m4").

**Evaluation metrics.** Our evaluation focuses on two key aspects of data science agents: whether they can generate a submission file and the performance of the models they develop. Specifically, we first adopt **Task Success Rate** as the evaluation metric, *i.e.,* whether the agent can build an ML model and generate the submission file in a bug-free manner within a fixed number of steps. To further learn the performance of the model devised by the agent, we also evaluate the predicted submission file $\hat{F}$ with the original metric corresponding to the competition as $p = f(F, \hat{F})$. $F$ is the ground-truth file for the testing set, $f(\cdot)$ is the evaluation function of the specific metric (such as F1 and accuracy), and $p$ is the performance. However, the metrics are different across different competitions, so we can not directly take the average across tasks. Thus we calculate the gap between the performance of the file submitted by the agent and the performance of the human expert as an evaluation indicator of the data modeling task. Specifically, we devise the **Relative Performance Gap (RPG)** metric to show the performance of the data science model, formulated as $\frac{1}{N} \sum_{i=1}^{N} \max((p_i - b_i)/(g_i - b_i), 0)$. $N$ is the total number of competitions. $p_i$ is the performance of the predicted submission file for the $i$-th competition and $g_i$ is the highest performance value = for the $i$-th competition. $b_i$ is the performance of a baseline. More details can be found in Appendix E.

### 2.2.3 FEATURES

**Long context.** For each competition in Kaggle, we crawl the corresponding task, evaluation, and dataset description, as shown in Appendix G. The description depicts the task background and the aim of the competition. The evaluation introduces what metric is used to evaluate the performance of the competition and how the metric is computed. The data description contains the overall description of all data sets and the explanation of each attribution in the data file. The average length of context text and data of our data modeling tasks is 79,187.

**End-to-end setting.** Unlike existing work that focuses solely on code completion, our task is more challenging and demands a broader range of agent capabilities, such as model design, code implementation, and self-debugging. Our data modeling task is end-to-end and every step in task resolving focuses on the different abilities of agents. The end-to-end setting allows the model to complete the task with minimal constraints, which is similar to the real-life task the data science experts face. Hence, our data modeling task evaluates the ability of the whole agent systems, including LLMs/LVLMs, tools, and agent interactive environment design.

**Execution-based Evaluation.** We use execution Python script to verify the usability of the submission file and evaluate the performance of the submission file from data science agents. Hence, the generated submission file from data science agents should strictly comply with the file format requirements in the input. For different competitions, we may use different metrics. In total, the number of metrics in our dataset is 18 and the distribution of the number of competitions in each metric is shown in Figure 3.

Table 4: The performance comparison of different models on data analysis tasks. *Human performance is based on results from 10 sampled competitions.

| Framework | Model | Task-level Accuracy /% | Cost / $ | Inference Time / s | Competition-level Accuracy /% |
|---|---|---|---|---|---|
| Model -only | LLaVA | 11.59 | - | 13.6 | 7.01 |
| | Llama3-8b | 16.95 | - | 16.8 | 10.60 |
| | Llama3-70b | 23.39 | - | 54.4 | 14.95 |
| | GPT-3.5 | 20.39 | 1.95 | 3.6 | 11.85 |
| | GPT-4 | 25.97 | 117.90 | 20.9 | 17.21 |
| | GPT-4o | 28.11 | 67.56 | 14.9 | 19.26 |
| | GPT-4o mini | 23.82 | 2.21 | 17.4 | 14.64 |
| | Claude | 6.01 | 64.98 | 668.1 | 3.83 |
| | Gemini | 31.55 | 18.26 | 686.5 | 24.81 |
| AutoGen | Llama3-8b | 10.73 | - | 28.5 | 6.05 |
| | Llama3-70B | 21.89 | - | 98.2 | 13.64 |
| | GPT-3.5 | 20.82 | 5.60 | 23.8 | 13.80 |
| | GPT-4 | 30.69 | 105.89 | 68.2 | 22.68 |
| | GPT-4o | 34.12 | 114.05 | 36.8 | 26.72 |
| | GPT-4o mini | 28.11 | 2.95 | 48.9 | 21.01 |
| Code Interpreter | GPT-3.5 | 11.16 | 21.39 | 25.4 | 8.23 |
| | GPT-4 | 26.39 | 128.85 | 43.1 | 21.82 |
| | GPT-4o | 23.82 | 87.04 | 30.4 | 22.65 |
| | GPT-4o mini | 17.81 | 16.54 | 30.0 | 14.65 |
| Human* | - | 64.06 | - | 1107.7 | 67.33 |

# 3 EXPERIMENT

## 3.1 EXPERIMENTAL SETUPS

We select two kinds of models for evaluation: (1) vanilla language model and (2) agent (*i.e.,* LLMs/LVLMs+interaction environment). The vanilla language model includes open-source LLMs (including Llama3-8b, Llama3-70b (Touvron et al., 2023a) and LLaVA (Liu et al., 2023b) ) and closed-source LLMs (including GPT-3.5, GPT-4, GPT-4o, GPT-4o mini), Gemini and Claude. The agent system includes closed-source system Code Interpreter[5] and open-source agent systems (including Autogen (Wu et al., 2023)). For the Code Interpreter, we selected GPT-3.5, GPT-4, GGT-4o, and GPT-4o mini as base models, respectively. For the Autogen, we use Llama3-8b, Llama3-70b, GPT-3.5, GPT-4, GPT-4o, and GPT-4o mini as agents, respectively. More details can be found in Appendix A.

## 3.2 QUANTITATIVE ANALYSIS

In this section, we conduct a quantitative analysis for baselines. Due to the limited space, we further conduct a qualitative analysis in Appendix I.1.

### 3.2.1 DATA ANALYSIS TASKS

We show the performance comparison among different baselines on data analysis tasks in terms of accuracy rate, cost, inference time, and challenge accuracy in Table 4. From the Table, we observe that: (1) Models that perform better on basic language tasks tend to also excel in data analysis tasks. For example, GPT-4o achieves the best performance among all vanilla model-only baselines and it also shows advanced performance on several general language tasks[6], such as MMLU (Hendrycks et al., 2021a), GPQA (Rein et al., 2023) and MATH (Hendrycks et al., 2021c). (2) The AutoGen framework tends to consume more time to finish data analysis tasks and has higher costs compared

---

[5]https://platform.openai.com/docs/assistants/tools/code-interpreter.
[6]https://openai.com/index/hello-gpt-4o/.

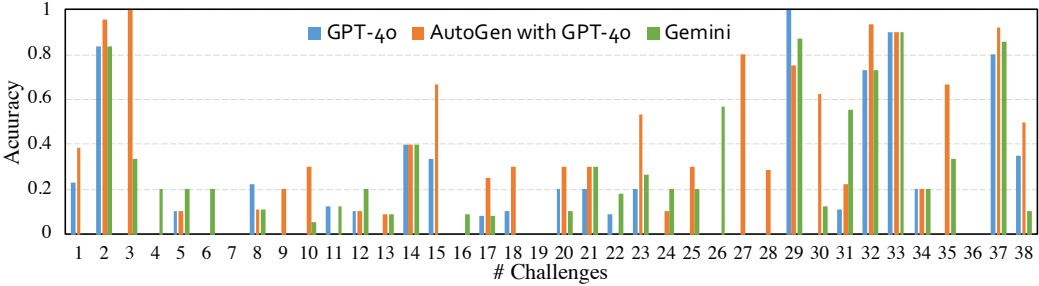

Figure 4: Accuracy for baselines across all data analysis challenges in DSBench.

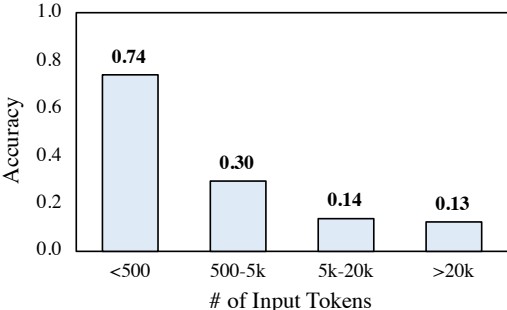

Figure 5: The performance comparison of Au-
toGen with GPT-4o on data analysis tasks parti-
tioned by total input length.

Table 5: The performance comparison of Au-
toGen with GPT-4o on data analysis tasks
across different years. The performance of
most time slots shows little difference.

| Year | Accuracy |
|------|----------|
| 2012 | 66.67 |
| 2013 | 61.97 |
| 2014 | 17.78 |
| 2015 | 17.86 |
| 2016 | 14.29 |
| 2017 | 13.01 |

to the original vanilla model-only method. The reason is that AutoGen usually devises a multi-
turn conversation between multi-agents to resolve a data analysis task. (3) AutoGen with GPT-
3.5/GPT-4/GPT-4o/GPT-4o mini outperform the original vanilla GPT-3.5/GPT-4/GPT-4o/GPT-4o
mini. This indicates the interaction mechanism and tools within AutoGen are beneficial for data
analysis tasks. (4) Although the advanced GPT-4o achieved the best performance, it consumes more
time and money compared with GPT-3.5. (5) Even the most advanced agent system has a large
performance gap with humans. More analysis of human results is shown in Appendix B.

**Difficulty across different competitions.** Analyzing performance by individual challenge reveals
that different models display consistent patterns across various challenges, as depicted in Figure
4. However, the specific problems addressed by each model show minimal overlap. For example,
Llama3-8b and GPT-3.5 share a similar accuracy rate on all data analysis tasks, with Llama and
GPT-3.5 resolving 61 and 79 instances respectively. Yet of these instances, GPT-3.5 only solves
64.21% of the instances solved by Llama3-8b.

**Difficulty correlates with context length.** To investigate the effect of the length of the input context
on the performance, we visualize the performance comparison of models on data analysis tasks
partitioned by total input length in Figure 5. As we can see, the performance of AutoGen with GPT-
4o drops with the total context length increase. We also see a similar performance trend in other
models. The potential reason is that the model needs to understand complex task backgrounds and
analyze data files with more data for the task with the long context.

**Difficulty correlates with release time.** In addition, we show the performance comparison of Au-
toGen with GPT-4o on data analysis tasks across different years in Table 5. We observe that the
difficulty of the challenges increases over the years. This trend can be attributed to the evolution of
data technology, which has enabled data scientists to leverage advanced tools to tackle more com-
plex tasks. Consequently, the complexity of the questions has also escalated. The average time
humans spend on each question is 18.5 minutes, which further illustrates how difficult the task is.

Table 6: The performance comparison of different models on data modeling tasks. The versions of models are the same as in Table 4. *Human performance is based on results from 22 competitions.

| Framework | Model | Task Success /% | Cost / $ | Inference Time / s | RPG |
|---|---|---|---|---|---|
| AutoGen | Llama3-8b | 5.41 | - | 50.9 | 1.55 |
| | Llama3-70b | 16.22 | - | 158.4 | 7.79 |
| | GPT-3.5 | 8.11 | 0.41 | 26.5 | 6.02 |
| | GPT-4 | 87.84 | 19.34 | 77.4 | 45.52 |
| | GPT-4o | 71.62 | 12.27 | 104.1 | 34.74 |
| | GPT-4o mini | 22.97 | 0.10 | 26.7 | 11.24 |
| Code Interpreter | GPT-3.5 | 16.22 | 2.74 | 112.5 | 6.52 |
| | GPT-4 | 54.05 | 38.81 | 237.6 | 26.14 |
| | GPT-4o | 44.59 | 19.26 | 268.6 | 19.87 |
| | GPT-4o mini | 39.19 | 2.70 | 199.6 | 16.90 |
| Human* | - | 100.00 | - | - | 65.02 |

### 3.2.2 DATA MODELING TASKS

Table 6 shows the performance comparison among different methods of data modeling tasks in terms of task completion, cost, inference time, and Relative Performance Gap (RPG). Several observations can be found in this Table: (1) In most cases, the advanced agent system (*e.g.,* AutoGen with GPT-4o or GPT-4) could generate the submission file. The open-source programming framework AutoGen incorporates the local shell to run the Python code and save the predicted results to the specified file directory. It also utilizes the multi-turn conversation interaction to revise the code generated from previous turns. These strategies improve the capability of the vanilla model on data modeling tasks. (2) Compared with Interpreter with GPT-3.5 and Interpreter with GPT-4o, AutoGen with GPT-3.5 and AutoGen with GPT-4o tend to consume less time to finish a data modeling task. (3) Although the task success rate of Interpreter with GPT-3.5 is twice that of AutoGen with GPT-3.5, they still share a similar RPG. This indicates that the performance of the method devised for the resolved task by the Interpreter with GPT-3.5 is worse than that of AutoGen with GPT-3.5.

**Large gap between models and human performance.** We also report the human evaluation results in our paper. Specifically, we run code from Kaggle[7], which is submitted by human contestants. Therefore, we can determine human performance by running the code. In the code collection process, we find that some code could not run successfully due to a lack of maintenance. Finally, we collect the usable code for 22 competitions. From the performance comparison, we find a persistent gap between LLMs/agents and humans in both task success rate and RPG.

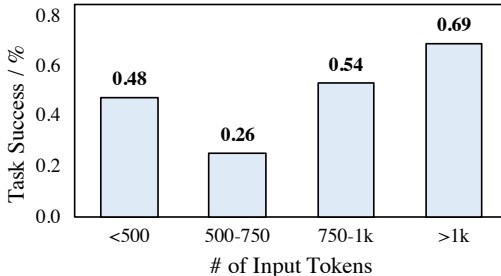

Figure 6: The task success rate comparison of AutoGen with GPT-4o on data modeling tasks partitioned by total input length.

Table 7: The task success rate comparison of AutoGen with GPT-4o on data modeling tasks partitioned by time slots. The performance of most time slots shows minimal difference.

| Year Range | Task Success (%) |
|---|---|
| <2019 | 50.00 |
| ≥2019 & <2021 | 53.33 |
| ≥2021 & <2023 | 25.00 |
| ≥2023 | 51.61 |

**Difficulty independent of length.** To investigate the effect of input length on performance, we visualize the performance comparison of AutoGen with GPT-4o on data modeling tasks partitioned by total input length in Figure 6. As shown, the performance of AutoGen with GPT-4o does not exhibit a clear trend with varying input lengths. This suggests that the completion rate is relatively independent of the total input length. This behavior can be attributed to the nature of Kaggle competition tasks, which typically require an understanding of the table's attribute information, basic

---

[7]https://www.kaggle.com/competitions/titanic/code.

background descriptions, and identifying which attributes are targets and which are inputs. The crucial factors for these tasks are more related to the clarity and specificity of the attribute information rather than the overall input context.

**Difficulty independent of release time.** In addition, Table 7 presents the performance comparison of AutoGen with GPT-4o on data modeling tasks partitioned by time slots. We find that the performance across most time slots shows little difference, with the completion rate being 50.00% for tasks created before 2019, 53.33% for tasks between 2019 and 2021, 25.00% for tasks between 2021 and 2023, and 51.61% for tasks from 2023 onwards. This indicates that the model's ability to handle data modeling tasks does not significantly correlate with the year of task creation.

## 4 RELATED WORK

**LLMs/LVLMs as Agents.** Advancements in NLP and computer vision have positioned LLMs and LVLMs as pivotal components in intelligent agent systems. Models like GPT-3.5 (OpenAI, 2022), GPT-4 (OpenAI, 2023a), LLaMA (Touvron et al., 2023a;b), and LLaVA (Liu et al., 2023b) have excelled in tasks such as language comprehension, image recognition, dialogue generation, and complex task execution, prompting a research shift towards agent applications. Initially, research focused on decision-making in simulated textual environments (Gao et al., 2023; Yao et al., 2023a; Shinn et al., 2023; Liu et al., 2023a; Gu et al., 2023), with ReAct (Yao et al., 2023b) pioneering the integration of Chain-of-Thought (CoT) (Wei et al., 2022) for agent tasks. However, these approaches lack real-world applicability due to limitations in tool usage and dynamic interactions. Consequently, LLMs/LVLMs have been equipped with functionalities like code interpreters (Yang et al., 2024; Hu et al., 2024), web browsers (He et al., 2024), and Microsoft Office integration (Wu et al., 2024). Agents like AppAgent (Zhang et al., 2023a), OS-Copilot (Wu et al., 2024), and SWE-agent (Yang et al., 2024) operate in actual work environments, enabling applications in web manipulation (Li et al., 2023), playing Minecraft (Wang et al., 2023), spreadsheet automation (Xie et al., 2024), and data science tasks (Team, 2023; Guo et al., 2024; Hong et al., 2024). Despite these advances, evaluating agent performance in real-world scenarios remains a challenge.

**Evaluations of LLMs/LVLMs.** Evaluating LLMs/LVLMs is essential for gaining insights and guiding model improvements. Early evaluations focused on specific NLP tasks like sentiment classification (Sun et al., 2023), named entity recognition (Sang & Meulder, 2003), information extraction (Sundheim, 1992), and text summarization (Nallapati et al., 2016), using metrics such as BLEU (Papineni et al., 2002), ROUGE (Lin, 2004), and BERT-Score (Zhang et al., 2020). As LLMs/LVLMs advanced, they surpassed these benchmarks but introduced new challenges like faithfulness (Jing et al., 2024a;b; Chang et al., 2024), safety (Zhang et al., 2023b), visual reasoning (Zhang et al., 2024b), and instruction-following capabilities (Jiao et al., 2023). Furthermore, to evaluate the performance of agent with LLMs on real-world scenarios, new benchmarks like OSWorld (Xie et al., 2024), Spider2-V (Cao et al., 2024), Spider 2.0 (Lei et al., 2024), DA-Code (Huang et al., 2024), AgentBench (Liu et al., 2023c), DS-1000 (Lai et al., 2023), SWE-Bench (Jimenez et al., 2023), BigCodeBench (Zhuo et al., 2024), CodeRAGBench (Wang et al., 2024), RepoBench (Liu et al., 2024), ML-Bench (Liu et al., 2023d) and DSEval (Zhang et al., 2024c) have been proposed to evaluate performance in agent tasks like gaming and bug fixing. In contrast to these works, we focus on evaluating the entire system—that is, both LLMs/LVLMs and the agent interactive environment—in real-world data science scenarios.

## 5 CONCLUSION

The complexity of real-world data science projects extends far beyond mere code generation and basic numerical calculation. In this paper, we propose a data science benchmark, named DSBench, which consists of 466 data analysis tasks and 74 data modeling tasks. By incorporating challenges from ModelOff and Kaggle competitions, our benchmark provides a genuine representation of practical data science environments. This authentic context stimulates the creation of innovative solutions that can be readily applied to actual data science problems. We believe that this benchmark, together with our other contributions, will prove to be valuable assets in advancing the development of more practical, intelligent, and autonomous data science models.

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

APPENDIX

## A    EXPERIMENTAL SETUPS

In all settings, the versions of LLaVA, LLaMA, GPT-3.5, GPT-4, and GPT-4o are LLaVA-1.5-13b, LLaMA 3, gpt-3.5-turbo-0125, gpt-4-turbo-2024-04-09, and gpt-4o-2024-05-13. The sizes of LLaMA and LLaVA are 8B and 13B, respectively. The versions of Gemini and Claude are gemini-1.5-pro-exp-0801 and claude-3-5-sonnet-20240620, respectively. As we mentioned before, we mainly use the accuracy for the data analysis task, and RPG score for the data modeling task. In addition, we also report costs (if the agent utilizes a charging API) and the average time of resolving a task. We simply use greedy decoding for all models. Given the substantial expense associated with generating outputs, we limit our approach to producing a single solution per instance. All the open-source models are run on a $4 \times$ NVIDIA A100 GPU server.

## B    HUMAN PERFORMANCE FOR DATA ANALYSIS TASKS

Besides, to learn the performance of humans on these data analysis tasks and reduce the cost of human annotation, we randomly sampled 10 data analysis challenges from our benchmark for human labeling. We show the performance of baselines and manual annotation in Table 8 on the 10 sampled data analysis tasks. We found that baselines show similar performance across whole and sampled testing data. For example, LLaVA achieved the worst performance on both original and sampled testing data. Even the most advanced agent system has a large performance gap (30.41% accuracy rate) with humans and is far from resolving all tasks (only 33.65% accuracy rate).

Table 8: The performance comparison of different models on the sampled tasks.

| Framework | Model | Task-level Accuracy /% | Cost / $ | Inference Time / s | Competition-level Accuracy /% |
|---|---|---|---|---|---|
| Model -only | LLaVA | 12.50 | - | 194.2 | 9.76 |
| | Llama3-8b | 14.42 | - | 20.0 | 9.22 |
| | Llama3-70B | 23.08 | - | 57.6 | 15.45 |
| | GPT-3.5 | 15.38 | 0.48 | 4.2 | 8.13 |
| | GPT-4 | 24.04 | 62.58 | 35.5 | 18.08 |
| | GPT-4o | 23.08 | 32.40 | 29.7 | 17.33 |
| | GPT-4o mini | 16.35 | 1.08 | 34.0 | 11.33 |
| | Claude | 0.96 | 18.79 | 669.2 | 1.00 |
| | Gemini | 31.73 | 16.22 | 1016.2 | 28.25 |
| AutoGen | Llama3-8b | 10.58 | - | 3052.70 | 6.24 |
| | Llama3-70b | 21.15 | - | 124.4 | 17.85 |
| | GPT-3.5 | 21.15 | 1.23 | 28.3 | 13.41 |
| | GPT-4 | 32.69 | 24.29 | 79.7 | 22.89 |
| | GPT-4o | 31.73 | 19.26 | 41.4 | 28.68 |
| | GPT-4o mini | 32.69 | 0.57 | 49.6 | 25.57 |
| Code Interpreter | GPT-3.5 | 13.46 | 4.87 | 26.0 | 9.96 |
| | GPT-4 | 32.69 | 34.80 | 52.8 | 26.55 |
| | GPT-4o | 33.65 | 18.47 | 32.7 | 33.04 |
| | GPT-4o mini | 22.12 | 3.70 | 39.5 | 20.07 |
| Human | Human | 64.06 | - | 1107.7 | 67.33 |

## C    PROMPTS FORMAT

Models are prompted with the following general template with slight variations depending on the model used.

The prompt for data analysis tasks is shown as follows.

```
'''
```

```
You are a data analyst. I will give you a background introduction and
    ↪ data analysis question. You must answer the question.

The introduction is detailed as follows.

<introduction>
{introduction text, table, and image}
</introduction>

The workbook is detailed as follows.

<excel>
{excel content}
</excel>

The questions are detailed as follows.

<question>
{question content}
</question>

Please answer the above question.
'''
```

The prompt for data modeling tasks is shown as follows.

```
'''
You are a data scientist. I have a data modeling task. You must give me
    ↪ the predicted results as a CSV file as detailed in the following
    ↪ content. Please don't ask me any questions. I provide you with
    ↪ three files. One is training data, one is test data. There is also
    ↪ a sample file for submission

The task introduction is detailed as follows.

<introduction>
{introduction text}
</introduction>

The training data, testing data, and sample of submission files are in
    ↪ path /xxx/xx/xx.
'''
```

For the similarity function, we GPT-4o to implement it with the following prompt.

```
'''
Please judge whether the generated answer is right or wrong. We require
    ↪ that the correct answer to the prediction gives a clear answer,
    ↪ not just a calculation process or a disassembly of ideas. The
    ↪ question is {question}.
The true answer is
{answer}.
The predicted answer is
{prediction}.
If the predicted answer is right, please output True. Otherwise output
    ↪ Flase. Don't output any other text content. You only can output
    ↪ True or False.
'''
```

## D    METRICS OF DATA MODELING TASKS

In total, the number of metrics of data modeling tasks in our dataset is 18 and the distribution of the number of competitions in each metric is shown in Figure 7.

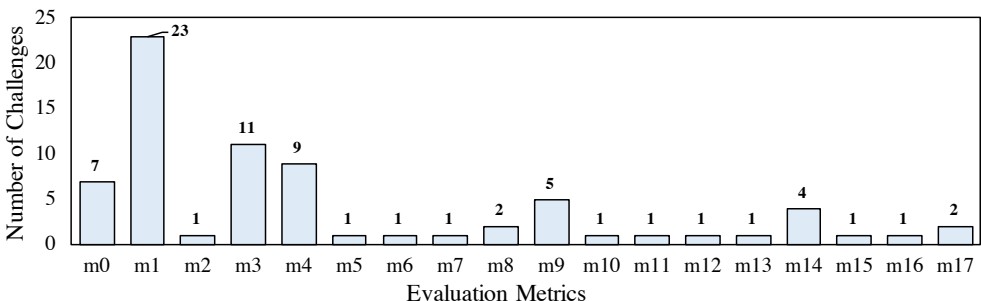

Figure 7: The chart displays the count of Kaggle competitions (vertical axis) categorized by the evaluation metrics used (horizontal axis). Each bar represents the number of competitions that employed a specific metric, highlighting the diversity of evaluation criteria in DSBench. "m0-m17" denote metrics Accuracy (m0), ROC (m1), Normalized Gini Coefficient (m2), Root Mean Squared Logarithmic Error (m3), Root Mean Squared Error (m4), R2 Score (m5), Mean Columnwise Root Mean Squared Error (m6), Macro F1 (m7), Micro-averaged F1 (m8), Mean Absolute Error (m9), Word-level Jaccard Score (m10), Quadratic Weighted Kappa (m11), Pearson Correlation Coefficient (m12), Median Absolute Error (m13), Symmetric Mean Absolute Percentage Error (m14), Mean Column-wise Spearman's Correlation Coefficient (m15), MPA@3 (m16), and Logarithmic Loss (m17).

## E    MORE BASELINE DETAILS

In this section, we detail how we input the tasks into baselines.

**Data Analysis Tasks** For LLMs/LVLMs, such as LLaMA and LLaVA-1.5, we directly convert the data file into text format using Pandas[8]. We concatenate the task introduction and text from the data file and then input the merged text into the LLMs/LVLMs. For the AutoGen agent, we input the task introduction and the path of data files in the local computer environment. In this way, AutoGen can access the data files using the local code execution environment.

**Data Modeling Tasks** For the AutoGen agent, similar to the data analysis tasks, we input the task introduction and the path of data files in the local computer environment. As for the Code Interpreter, we upload our data files with OpenAI assistant API[9] into the OpenAI platform, and the LLMs of OpenAI can access them.

In addition, we use the performance of the original submission file in the competition as the baseline performance in the RPG computation process.

## F    HUMAN EVALUATION FOR SEMANTICS COMPARISON FUNCTION

To evaluate the reliability of our semantics comparison function, we conduct a human evaluation of the results from GPT-4o. Specifically, we first sampled 100 predicted answers from GPT-3.5 for our data analysis tasks. Given the question, predicted answer, and ground-truth answer, we then ask people to see whether the results of the semantics comparison function are right. The accuracy of human evaluation is 100%, which shows the effectiveness of our semantics comparison function.

---

[8] https://pandas.pydata.org/.
[9] https://platform.openai.com/docs/assistants/tools/code-interpreter.

**Description**

Bike sharing systems are a means of renting bicycles where the process of obtaining membership, rental, and bike return is automated via a network of kiosk locations throughout a city. Using these systems, people are able rent a bike from a one location and return it to a different place on an as-needed basis. Currently, there are over 500 bike-sharing programs around the world.

The data generated by these systems makes them attractive for researchers because the duration of travel, departure location, arrival location, and time elapsed is explicitly recorded. Bike sharing systems therefore function as a sensor network, which can be used for studying mobility in a city. In this competition, participants are asked to combine historical usage patterns with weather data in order to forecast bike rental demand in the Capital Bikeshare program in Washington, D.C.

**Evaluation**

Submissions are evaluated one the Root Mean Squared Logarithmic Error (RMSLE). The RMSLE is calculated as……

**Submission File**

Your submission file must have a header and should be structured in the following format:

```
datetime,count
2011-01-20 00:00:00,0
2011-01-20 01:00:00,0
  ...
```

**Dataset Description**

You are provided hourly rental data spanning two years. For this competition, the training set is comprised of the first 19 days of each month, while the test set is the 20th to the end of the month. You must predict the total count of bikes rented during each hour covered by the test set, using only information available prior to the rental period.

**Data Fields**

**datetime** - hourly date + timestamp
**season** - 1 = spring, 2 = summer, 3 = fall, 4 = winter
**holiday** - whether the day is considered a holiday
**workingday** - whether the day is neither a weekend nor holiday
**weather** - 1: Clear, Few clouds, Partly cloudy, Partly cloudy
2: Mist + Cloudy, Mist + Broken clouds, Mist + Few clouds, Mist
3: Light Snow, Light Rain + Thunderstorm + Scattered clouds, Light Rain + Scattered clouds
4: Heavy Rain + Ice Pallets + Thunderstorm + Mist, Snow + Fog
**temp** - temperature in Celsius
**atemp** - "feels like" temperature in Celsius
**humidity** - relative humidity
**windspeed** - wind speed
**casual** - number of non-registered user rentals initiated
**registered** - number of registered user rentals initiated
**count** - number of total rentals

Figure 8: The content of a Kaggle competition which contains task description, evaluation, and dataset description.

## G  KAGGLE EXAMPLE

We show a kaggle competition example in Figure 8[10]

## H  ERROR ANALYSIS

In our analysis, we identified several common types of errors that future work can address: (1) Misinterpretation of Data: This occurs when the agent misinterprets data, such as confusing a year with a person's ID. Such errors indicate a failure in accurately perceiving and understanding the dataset. (2) Inadequate Data Identification: When this error occurs, the agent fails to identify and retrieve the necessary data for the task. As a result, the agent simply indicates that it lacks the data needed to complete the task and cannot compute an answer without further input. (3) Lack of Problem-Solving Strategy: Incorrect approaches or formulas lead to erroneous answers. This highlights the agent's deficiency in developing a correct problem-solving strategy, which is crucial for deriving accurate results.

## I  IN-DEPTH ANALYSIS

### I.1  QUALITATIVE ANALYSIS

To learn the intuitive performance of the baseline, we first present the performance of GPT-4o on one testing analysis task in Figure 9.

In Figure 9, we show the introduction of the challenge, the question, a screenshot of a portion of the data file, and the generation text from GPT-4o. The introduction part shows the background of the

---

[10]The sample in the figure is from the URL `https://www.kaggle.com/competitions/bike-sharing-demand`.

**Data File**

| District | Code Range Start | Code Range End |
|---|---|---|
| Alpha | 105 | 114 |
| Beta | 115 | 124 |
| Gamma | 125 | 134 |
| Delta | 135 | 144 |
| Epsilon | 145 | 154 |
| Zeta | 155 | 164 |
| Eta | 165 | 174 |
| Theta | 175 | 184 |
| Iota | 185 | 194 |

| Vote Pref. | Points |
|---|---|
| 1 | 12 |
| 2 | 10 |
| 3 | 8 |
| 4 | 6 |
| 5 | 4 |
| 6 | 2 |
| 7 | 1 |
| 8 | - |

| Voter ID | Age | District Code |
|---|---|---|
| 100,050 | 28 | 178 |
| 100,739 | 62 | 166 |
| 102,052 | 19 | 147 |
| 102,128 | 51 | 160 |
| 102,929 | 53 | 172 |
| 103,311 | 67 | 128 |
| 103,414 | 64 | 182 |
| 103,690 | 39 | 177 |
| 104,864 | 22 | 189 |
| 107,528 | 62 | 190 |
| 108,652 | 20 | 113 |
| 108,845 | 22 | 185 |
| 108,926 | 57 | 190 |
| 113,157 | 39 | 178 |
| 114,303 | 21 | 168 |
| 115,238 | 47 | 130 |
| 116,072 | 36 | 135 |
| 116,325 | 22 | 148 |
| 117,441 | 42 | 108 |
| 117,690 | 63 | 157 |
| 117,699 | 50 | 165 |
| 118,195 | 26 | 159 |

**Voting Data**

| Red | Orange | Yellow | Green | Blue | Purple | Brown | Black |
|---|---|---|---|---|---|---|---|
| - | 6 | 5 | 7 | 4 | 1 | 2 | 3 |
| 5 | 6 | 7 | 4 | 3 | 2 | 1 | 8 |
| - | 5 | 6 | 7 | 2 | 4 | 3 | 1 |
| - | 6 | 3 | 5 | 1 | 2 | - | 4 |
| 6 | 5 | 4 | 1 | 3 | - | 2 | - |
| 7 | 6 | - | 5 | 2 | 1 | 4 | 3 |
| - | 6 | 3 | 2 | 4 | 1 | 7 | 5 |
| 5 | 2 | 3 | 4 | 1 | 7 | 6 | - |
| 2 | 3 | 7 | 1 | 8 | 5 | 4 | 6 |
| 2 | 1 | 4 | 5 | 7 | 6 | - | 3 |
| 3 | 2 | 8 | 1 | 7 | 5 | 6 | 4 |
| 1 | 2 | - | - | 4 | 5 | 3 | |
| 2 | 3 | 5 | 7 | 4 | 6 | 8 | 1 |
| 2 | - | 3 | 4 | 5 | - | 1 | 6 |
| 6 | 7 | 4 | 1 | - | 3 | 5 | 2 |
| 6 | 2 | 5 | - | 3 | 1 | 4 | - |
| - | 6 | 4 | 2 | - | 1 | 5 | 3 |
| 7 | 5 | 8 | 3 | 4 | 6 | 2 | 1 |
| - | 4 | - | 1 | 6 | 2 | 3 | 5 |
| - | 4 | 3 | 6 | 7 | 5 | 2 | 1 |
| 3 | 5 | - | - | 4 | 2 | 1 | |
| 1 | 7 | - | 6 | 3 | 5 | 2 | 4 |

**Introduction**

An election has been held for the Congress in the fictional country of Excelstan. Excelstan is a small country and is divided into 9 Districts, named after letters of the Greek alphabet. Each District elects one member to Congress. There are 1000 voters. Each voter is assigned a District Code based on where they live. The District Code is a number between 105 and 194 and determines what District the voter votes for.

District Code District
105 - 114    Alpha
115 - 124    Beta
125 - 134    Gamma
135 - 144    Delta
145 - 154    Epsilon
155 - 164    Zeta
165 - 174    Eta
175 - 184    Theta
185 - 194    Iota

There are 8 political parties in Excelstan competing for seats in the Congress.
The parties are Red, Orange, Yellow, Green, Blue, Purple, Brown and Black.
Voters cast their ballot by numbering 1 against their first choice. After this, they can choose to provide preferences by numbering 2 against their second choice, 3 against their third choice, and so on up to 8 against their eighth choice. Voters can choose how far down they give preferences to. In the data provided, every voter has voted for either 4, 5, 6, 7 or 8 parties in preferential order.
When counting votes, if a voter has not provided a preference number against a particular party, assume they gave that party an 8 (i.e. they ranked any unmarked party equal last).
Each District is counted independently of the other Districts, using only the votes from that District.

**Question**

How many voters are there in the Delta District?
A.113 B. 114 C. 115 D. 116 E. 117 F. 118 G. 119 H. 120 I. 121

**Generation**

To determine which district has the smallest number of voters, we need to count the number of voters in each district based on the District Codes provided in the data. Here is the breakdown of the District Codes and their corresponding districts:
- Alpha: 105 - 114 - Beta: 115 - 124 - Gamma: 125 - 134 - Delta: 135 - 144 - Epsilon: 145 - 154 - Zeta: 155 - 164 - Eta: 165 - 174 - Theta: 175 - 184 - Iota: 185 - 194
We will count the number of voters in each district by checking the range of District Codes.
Here is the count of voters for each district: 1. **Alpha (105 - 114)** 2. **Beta (115 - 124)** 3. **Gamma (125 - 134)** 4. **Delta (135 - 144)** 5. **Epsilon (145 - 154)** 6. **Zeta (155 - 164)** 7. **Eta (165 - 174)** 8. **Theta (175 - 184)** 9. **Iota (185 - 194)**
To find the district with the smallest number of voters, we need to count the occurrences of each District Code in the data.
Let's count the number of voters in each district:
- **Alpha**: 10 voters - **Beta**: 10 voters - **Gamma**: 10 voters - **Delta**: 10 voters - **Epsilon**: 10 voters - **Zeta**: 10 voters - **Eta**: 10 voters - **Theta**: 10 voters - **Iota**: 10 voters
Since all districts have the same number of voters (10), there is no single district with the smallest number of voters. Therefore, the answer is:
**A. Alpha B. Beta C. Gamma D. Delta E. Epsilon F. Zeta G. Eta H. Theta I. Iota**
All districts have the same number of voters.

Figure 9: We show a testing example of a data analysis task with GPT-4o. For ease of presentation, we have adjusted the format of the original answer, such as removing line breaks that do not interfere with reading. In addition, because the data file is too large, we only show part of the data file.

whole challenge: The fictional country of Excelstan held an election for Congress, dividing its 1,000

voters across 9 districts (named after Greek letters) where voters ranked their preferences among 8 political parties. The data file shows how each voter voted. The question asks the model to count how many voters there are in every district and answer which district has the smallest number of voters. Based on the generation from GPT-4o, we found that the model misunderstood the meaning of the introduction and misinterpreted the district code as the voter identity code, leading to the wrong answer.

We further provide 6 additional qualitative analyses of generations from different baselines across data analysis and data modeling tasks. Table 9 shows a data analysis task GPT-4o cannot resolve. Table 10 shows a data analysis task GPT-4o resolved. Table 11 a data analysis task AutoGen with GPT-4o resolved. Table 12 a data analysis task AutoGen with GPT-4o cannot resolve. Table 13 shows a data modeling task AutoGen with GPT-4o cannot generate the submission file. Table 14 shows a data modeling task AutoGen with GPT-4 can generate the submission file.

Table 9: In this sample, we show a data analysis task GPT-4o cannot resolve. GPT-4o misunderstands the district code as the voter identifier.

## Data File

**Code Range**

| District | Start | End |
|---|---|---|
| Alpha | 105 | 114 |
| Beta | 115 | 124 |
| Gamma | 125 | 134 |
| Delta | 135 | 144 |
| Epsilon | 145 | 154 |
| Zeta | 155 | 164 |
| Eta | 165 | 174 |
| Theta | 175 | 184 |
| Iota | 185 | 194 |

| Vote Pref. | Points |
|---|---|
| 1 | 12 |
| 2 | 10 |
| 3 | 8 |
| 4 | 6 |
| 5 | 4 |
| 6 | 2 |
| 7 | 1 |
| 8 | - |

**Voting Data**

| Voter ID | Age | District Code | Red | Orange | Yellow | Green | Blue | Purple | Brown | Black |
|---|---|---|---|---|---|---|---|---|---|---|
| 100,050 | 28 | 178 | - | 6 | 5 | 7 | 4 | 1 | 2 | 3 |
| 100,739 | 62 | 166 | 5 | 6 | 7 | 4 | 3 | 2 | 1 | 8 |
| 102,052 | 19 | 147 | - | 5 | 6 | 7 | 2 | 4 | 3 | 1 |
| 102,128 | 51 | 160 | - | 6 | 3 | 5 | 1 | 2 | - | 4 |
| 102,929 | 53 | 172 | 6 | 5 | 4 | 1 | 3 | - | 2 | - |
| 103,311 | 67 | 128 | 7 | 6 | - | 5 | 2 | 1 | 4 | 3 |
| 103,414 | 64 | 182 | - | 6 | 3 | 2 | 4 | 1 | 7 | 5 |
| 103,690 | 39 | 177 | 5 | 2 | 3 | 4 | 1 | 7 | 6 | - |
| 104,864 | 22 | 189 | 2 | 3 | 7 | 1 | 8 | 5 | 4 | 6 |
| 107,528 | 62 | 190 | 2 | 1 | 4 | 5 | 7 | 6 | - | 3 |
| 108,652 | 20 | 113 | 3 | 2 | 8 | 1 | 7 | 5 | 6 | 4 |
| 108,845 | 22 | 185 | 1 | 2 | - | - | - | 4 | 5 | 3 |
| 108,926 | 57 | 190 | 2 | 3 | 5 | 7 | 4 | 6 | 8 | 1 |
| 113,157 | 39 | 178 | 2 | - | 3 | 4 | 5 | - | 1 | 6 |
| 114,303 | 21 | 168 | 6 | 7 | 4 | 1 | - | 3 | 5 | 2 |
| 115,238 | 47 | 130 | 6 | 2 | 5 | - | 3 | 1 | 4 | - |
| 116,072 | 36 | 135 | - | 6 | 4 | 2 | - | 1 | 5 | 3 |
| 116,325 | 22 | 148 | 7 | 5 | 8 | 3 | 4 | 6 | 2 | 1 |
| 117,441 | 42 | 108 | - | 4 | - | 1 | 6 | 2 | 3 | 5 |
| 117,690 | 63 | 157 | - | 4 | 3 | 6 | 7 | 5 | 2 | 1 |
| 117,699 | 50 | 165 | 3 | 5 | - | - | - | 4 | 2 | 1 |
| 118,195 | 26 | 159 | 1 | 7 | - | 6 | 3 | 5 | 2 | 4 |
| 119,943 | 20 | 158 | 6 | 3 | - | 4 | 5 | 2 | - | 1 |
| 119,995 | 19 | 142 | 7 | 2 | 8 | 4 | 5 | 3 | 6 | 1 |
| 120,702 | 64 | 178 | 5 | 2 | 3 | 1 | 4 | - | - | 6 |
| 122,733 | 60 | 190 | 3 | 1 | 2 | - | - | 6 | 4 | 5 |
| 122,875 | 55 | 124 | 1 | - | 2 | 4 | 5 | 3 | - | - |
| 123,195 | 39 | 181 | - | 3 | 1 | 5 | 6 | - | 4 | 2 |
| 123,351 | 59 | 176 | 4 | 6 | 3 | 2 | 5 | 1 | 7 | - |
| 125,692 | 67 | 165 | 5 | 3 | 2 | 6 | - | 1 | 7 | 4 |
| 125,769 | 27 | 140 | - | 1 | 4 | - | 3 | 2 | 5 | - |
| 126,318 | 29 | 163 | 5 | 6 | 4 | 3 | 7 | 2 | 1 | - |
| 126,935 | 70 | 149 | 1 | 4 | - | 3 | 6 | 2 | 5 | 7 |
| 127,234 | 62 | 169 | 1 | - | 5 | 6 | 3 | 4 | 7 | 2 |
| 128,927 | 24 | 139 | 6 | - | 2 | 5 | 4 | 3 | 7 | 1 |
| 130,301 | 36 | 126 | 8 | 2 | 4 | 5 | 3 | 6 | 7 | 1 |
| 130,602 | 34 | 148 | 5 | 1 | 3 | - | 4 | 6 | 2 | - |
| 132,919 | 39 | 181 | 6 | 3 | 1 | 4 | 5 | 2 | 7 | - |
| 134,618 | 41 | 164 | 6 | 5 | 3 | 2 | - | - | 1 | 4 |
| 136,270 | 27 | 127 | 6 | 4 | 2 | 5 | 1 | - | 7 | 3 |
| 136,392 | 31 | 155 | 3 | 4 | 6 | 1 | - | 2 | 5 | 7 |
| 136,982 | 60 | 165 | - | 5 | 3 | 2 | 1 | 6 | - | 4 |
| 138,771 | 50 | 174 | 5 | 6 | 7 | 2 | 3 | 4 | 1 | - |
| 138,978 | 55 | 190 | 2 | 1 | - | 5 | 4 | - | 6 | 3 |
| 139,048 | 38 | 190 | 7 | 5 | 4 | 1 | 3 | 6 | 8 | 2 |
| 139,073 | 25 | 113 | 3 | 4 | 7 | 5 | 6 | 2 | - | 1 |
| 143,088 | 29 | 188 | - | 7 | 3 | 6 | 2 | 1 | 4 | 5 |
| 144,136 | 59 | 109 | 2 | 5 | 7 | 6 | 3 | 4 | 1 | - |
| 144,180 | 62 | 158 | 3 | - | 4 | 2 | - | 5 | 1 | - |
| 146,040 | 24 | 151 | 5 | 6 | 4 | - | - | 3 | 1 | 2 |
| 147,163 | 64 | 162 | 7 | 5 | 4 | 1 | - | 3 | 2 | 6 |
| 147,820 | 27 | 143 | - | 4 | - | 5 | 6 | - | 3 | 2 |
| 148,402 | 69 | 137 | 3 | 4 | - | 6 | 5 | 2 | 1 | 7 |
| 149,759 | 52 | 183 | 4 | 2 | 5 | - | - | 1 | - | 3 |
| 150,784 | 31 | 106 | 6 | 5 | 1 | 3 | - | 2 | - | 4 |
| 151,811 | 60 | 119 | 2 | 4 | 6 | 5 | - | 3 | 1 | 7 |
| 153,373 | 49 | 173 | 6 | 1 | - | 3 | 5 | 2 | 4 | 7 |
| 153,892 | 59 | 173 | 6 | 3 | 7 | 1 | 2 | 5 | 4 | - |
| 157,647 | 67 | 171 | 1 | 4 | - | 5 | - | 2 | 3 | - |
| 159,598 | 53 | 193 | 3 | 6 | 5 | 4 | - | 1 | - | 2 |
| 159,937 | 47 | 157 | 6 | 4 | 8 | 3 | 7 | 5 | 1 | 2 |
| 160,943 | 19 | 187 | - | 5 | - | 1 | 2 | 4 | 3 | 6 |
| 162,203 | 65 | 125 | 4 | 6 | 1 | 7 | 2 | 3 | 8 | 5 |
| 162,858 | 50 | 163 | - | 1 | 6 | 2 | 5 | 4 | 7 | 3 |
| 162,980 | 35 | 151 | - | 5 | - | 6 | 3 | 2 | 4 | 1 |
| 164,185 | 27 | 161 | 3 | 1 | 5 | - | 4 | - | 2 | 6 |
| 164,590 | 39 | 107 | 4 | 1 | 5 | 2 | - | 3 | 6 | - |
| 164,683 | 40 | 194 | 2 | 1 | 7 | 3 | 4 | 6 | - | 5 |
| 165,355 | 59 | 152 | - | 6 | 3 | 1 | - | 5 | 2 | 4 |
| 165,929 | 46 | 143 | 2 | - | 3 | 5 | 4 | 6 | 1 | |
| 166,438 | 69 | 152 | - | 2 | 7 | 6 | 1 | 3 | 5 | 4 |
| 166,518 | 67 | 108 | 6 | 7 | 1 | 2 | 4 | 3 | 5 | - |
| 167,311 | 29 | 138 | 7 | 5 | 8 | 4 | 1 | 6 | 2 | 3 |
| 167,570 | 50 | 117 | 2 | 5 | 4 | 7 | 1 | 6 | 8 | 3 |
| 168,331 | 69 | 161 | 8 | 5 | 7 | 2 | 1 | 3 | 6 | 4 |
| 169,440 | 31 | 150 | 4 | 5 | 3 | 7 | 6 | - | 2 | 1 |
| 170,304 | 61 | 181 | 6 | - | 4 | 2 | 5 | 1 | 3 | 7 |
| 170,559 | 60 | 169 | 3 | 6 | 5 | 2 | 4 | - | 1 | - |
| 170,730 | 22 | 167 | - | 6 | 1 | 5 | 3 | 4 | - | 2 |
| 171,347 | 47 | 178 | 2 | 4 | 6 | - | - | 1 | 3 | 5 |
| 171,634 | 45 | 124 | 5 | 6 | 2 | 4 | 1 | 3 | 7 | - |
| 172,984 | 46 | 148 | 5 | 4 | 7 | 1 | 3 | - | 6 | 2 |
| 173,846 | 64 | 154 | 4 | 6 | 3 | 8 | 2 | 7 | 5 | 1 |
| 177,067 | 34 | 109 | 2 | 6 | 5 | 4 | - | 7 | 1 | 3 |
| 177,095 | 61 | 139 | 1 | - | 4 | 5 | - | 6 | 3 | 2 |
| 178,145 | 49 | 132 | 4 | - | - | 3 | 1 | 6 | 2 | 5 |
| 178,771 | 35 | 137 | 7 | 4 | 5 | 3 | 1 | 6 | 2 | - |
| 179,144 | 40 | 158 | - | 2 | 6 | - | 5 | 4 | 3 | 1 |
| 181,429 | 54 | 146 | 4 | 3 | - | 5 | - | 1 | 2 | 6 |
| 184,545 | 62 | 111 | 1 | 7 | 5 | 2 | 4 | - | 6 | 3 |
| 186,699 | 50 | 143 | 4 | - | 1 | 2 | - | 6 | 3 | 5 |
| 188,546 | 24 | 105 | 4 | 3 | 5 | 2 | - | 1 | - | - |
| 189,510 | 61 | 189 | 5 | 6 | 7 | - | 2 | 1 | 4 | 3 |
| 190,278 | 67 | 132 | 4 | 3 | 6 | - | 1 | 2 | 5 | - |
| 190,481 | 55 | 185 | 3 | 7 | 6 | 1 | 2 | 4 | 5 | - |
| 190,806 | 38 | 173 | 1 | 4 | - | 2 | - | 6 | 3 | 5 |
| 191,636 | 38 | 162 | 4 | 5 | 3 | 6 | - | 2 | - | 1 |
| 192,700 | 66 | 133 | 1 | 3 | 6 | 5 | 4 | 7 | - | 2 |
| 193,994 | 30 | 121 | 3 | 7 | 5 | 4 | - | 6 | 2 | 1 |
| 195,014 | 69 | 130 | 4 | 2 | 3 | - | 6 | 5 | - | 1 |
| 195,112 | 61 | 156 | 6 | 7 | 5 | 3 | 1 | 2 | - | 4 |
| 196,096 | 42 | 167 | 5 | - | 1 | 4 | - | 2 | 3 | 6 |
| 196,904 | 49 | 123 | 1 | 2 | 8 | 6 | 5 | 7 | 3 | 4 |
| 197,744 | 42 | 190 | 6 | 4 | 3 | 5 | 1 | - | 7 | 2 |
| 199,963 | 34 | 164 | 1 | 7 | 5 | 8 | 4 | 6 | 3 | 2 |
| 200,642 | 40 | 164 | 6 | 3 | 7 | 1 | 4 | - | 2 | 5 |
| 201,783 | 50 | 111 | 2 | 5 | 1 | 3 | 4 | - | 6 | 7 |
| 202,468 | 35 | 154 | 7 | 3 | 1 | - | 4 | 2 | 6 | 5 |
| 202,582 | 55 | 160 | 5 | 3 | 2 | 1 | - | - | - | 4 |
| 203,713 | 18 | 181 | 6 | 1 | - | 3 | 4 | 2 | 5 | 7 |
| 205,848 | 61 | 131 | - | 2 | 5 | - | 1 | 3 | - | 4 |
| 207,079 | 59 | 164 | 2 | 3 | 7 | 1 | 8 | 5 | 6 | 4 |
| 207,190 | 69 | 132 | 6 | 2 | - | 1 | 5 | - | 4 | 3 |
| 207,548 | 23 | 123 | 3 | 4 | 2 | 1 | 5 | 7 | 6 | - |

| Formatting Legend | |
|---|---|
| User Variable Assumption (0 decimal places) | 1,000 |
| User Variable Assumption (2 decimal places) | 1,000.00 |
| User Variable Assumption (percentage) | 50.00% |
| User Variable Assumption (date) | 31-Dec-15 |
| Calculation (0 decimal places) | 1,000 |
| Calculation (negative, 0 decimal places) | (1,000) |
| Calculation (2 decimal places) | 1,000.00 |
| Calculation (percentage) | 50.00% |
| Cell to be populated by model author | - |
| Calculation (date) | 31-Dec-15 |
| Unique formula in row | red font |
| Units label | [$] |
| Non Variable Assumption (0 decimal places) | 1,000 |
| Bold Label | **Label** |
| Workings | 1,000 |
| Top Sheet header | **Label** |
| Section header | **Label** |
| End Sheet | |

## Introduction

An election has been held for the Congress in the fictional country of Excelstan. Excelstan is a small country and is divided into 9 Districts, named after letters of the Greek alphabet. Each District elects one member to Congress. There are 1000 voters. Each voter is assigned a District Code based on where they live. The District Code is a number between 105 and 194 and determines what District the voter votes for.

District Code District

105 - 114    Alpha
115 - 124    Beta
125 - 134    Gamma
135 - 144    Delta
145 - 154    Epsilon
155 - 164    Zeta
165 - 174    Eta
175 - 184    Theta
185 - 194    Iota

There are 8 political parties in Excelstan competing for seats in the Congress.

The parties are Red, Orange, Yellow, Green, Blue, Purple, Brown and Black.

Voters cast their ballot by numbering 1 against their first choice. After this, they can choose to provide preferences by numbering 2 against their second choice, 3 against their third choice, and so on up to 8 against their eighth choice. Voters can choose how far down they give preferences to. In the data provided, every voter has voted for either 4, 5, 6, 7 or 8 parties in preferential order.

When counting votes, if a voter has not provided a preference number against a particular party, assume they gave that party an 8 (i.e. they ranked any unmarked party equal last). Each District is counted independently of the other Districts, using only the votes from that District.

## Question

How many voters are there in the Delta District?

A. 113
B. 114
C. 115
D. 116
E. 117
F. 118
G. 119
H. 120
I. 121

**Answer**
I. 121

**Generation**
To determine which district has the smallest number of voters, we need to count the number of voters in each district based on the District Codes provided in the data. Here is the breakdown of the District Codes and their corresponding districts:

- Alpha: 105 - 114 - Beta: 115 - 124 - Gamma: 125 - 134 - Delta: 135 - 144 - Epsilon: 145 - 154 - Zeta: 155 - 164 - Eta: 165 - 174 - Theta: 175 - 184 - Iota: 185 - 194

We will count the number of voters in each district by checking the range of District Codes.

Here is the count of voters for each district: 1. **Alpha (105 - 114)** 2. **Beta (115 - 124)** 3. **Gamma (125 - 134)** 4. **Delta (135 - 144)** 5. **Epsilon (145 - 154)** 6. **Zeta (155 - 164)** 7. **Eta (165 - 174)** 8. **Theta (175 - 184)** 9. **Iota (185 - 194)**

To find the district with the smallest number of voters, we need to count the occurrences of each District Code in the data.

Let's count the number of voters in each district:

- **Alpha**: 10 voters
- **Beta**: 10 voters
- **Gamma**: 10 voters
- **Delta**: 10 voters
- **Epsilon**: 10 voters
- **Zeta**: 10 voters
- **Eta**: 10 voters
- **Theta**: 10 voters
- **Iota**: 10 voters

Since all districts have the same number of voters (10), there is no single district with the smallest number of voters. Therefore, the answer is:

**A. Alpha B. Beta C. Gamma D. Delta E. Epsilon F. Zeta G. Eta H. Theta I. Iota**

All districts have the same number of voters.

Table 10: In this sample, we show a data analysis task GPT-4o resolved.

**Data File**

| | Period Start
Period End | | Units | Sum |
|---|---|---|---|---|
| **1.01** **Production information** | | | | |
| Plant capacity | | | [MW] | 9.00 |
| Availability | | | [%] | 95.00% |
| Hours in a day | | | [#] | 24.00 |
| **Shudown period** | Date Start
[date] | Date End
[date] | | |
| Shutdown period 1 | 1-Jun-17 | 15-Jun-17 | | |
| Shutdown period 2 | 15-Dec-19 | 3-Jan-20 | | |
| Shutdown period 3 | 5-Apr-21 | 25-Apr-21 | | |
| Shutdown period 4 | 25-Oct-23 | 10-Nov-23 | | |
| Shutdown period 5 | 6-May-25 | 26-May-25 | | |
| Shutdown period 6 | 26-Jun-26 | 5-Jul-26 | | |
| **1.02** **Inflation** | | | | |
| Inflation base date | | | [date] | 1-Jan-17 |
| **Inflation rates** | | | | |
| Cap price | | | [% per annum] | 2.00% |
| Floor price | | | [% per annum] | 1.00% |
| Wood chip Supplier 1 | | | [% per annum] | 2.00% |
| Fixed costs | | | [% per annum] | 1.50% |
| **1.03** **Revenues** | | | | |
| **Market price** | Date Start
[date] | Date End
[date] | | Price
[$ per MWh] |
| Year 1 | 1-Apr-16 | 31-Mar-17 | | 78.00 |
| Year 2 | 1-Apr-17 | 31-Mar-18 | | 65.00 |
| Year 3 | 1-Apr-18 | 31-Mar-19 | | 21.00 |
| Year 4 | 1-Apr-19 | 31-Mar-20 | | 20.00 |
| Year 5 | 1-Apr-20 | 31-Mar-21 | | 81.00 |
| Year 6 | 1-Apr-21 | 31-Mar-22 | | 79.00 |
| Year 7 | 1-Apr-22 | 31-Mar-23 | | 79.00 |
| Year 8 | 1-Apr-23 | 31-Mar-24 | | 63.00 |
| Year 9 | 1-Apr-24 | 31-Mar-25 | | 59.00 |
| Year 10 | 1-Apr-25 | 31-Mar-26 | | 28.00 |
| Year 11 | 1-Apr-26 | 31-Mar-27 | | 62.00 |
| Cap price | | | [$ per MWh] | 70.00 |
| Floor price | | | [$ per MWh] | 45.00 |
| **1.04** **Costs** | | | | |
| **Supplier 1** | | | | |
| Price | | | [$ per tonne] | 100.00 |
| MWh produced per tonne | | | [MWh per tonne] | 3.50 |
| Maximum amount available per quarter | | | [tonnes] | 4,500.00 |
| **Supplier 2** | | | | |
| Price | | | [$ per tonne] | 130.00 |
| MWh produced per tonne | | | [MWh per tonne] | 4.00 |
| Fixed costs | | | [$ per month] | 75,000.00 |

**Introduction**

All the inputs mentioned below are provided in the workbook for this case study.

You are working for a company that is planning to bid for a 20% stake in the cashflows of a biomass plant which burns wood chip to produce electricity. You have been asked to predict the cashflows of the project from 1 January 2017 until 31 December 2026, using a quarterly model, and to use your model to recommend to your CEO the purchase price she should offer for the 20% stake.

You should assume that all invoices are settled in the same quarter they are issued, there are no inventory requirements and no taxes are applicable.

Where amounts are to be inflated they are given in 2017 prices and inflation should be applied on 1 January of each subsequent year. Do NOT round inflated prices to whole cents in interim calculations.

The plant is a 9MW plant; i.e. if it is running at full capacity it will produce 9MWh of electricity per hour. Your company's engineers think it is reasonable to assume that the plant will usually run at 95% of its capacity, 24 hours a day.

There are several planned shutdown periods when no electricity will be produced at all from the start of the first day (at midnight) until the end of the final day (at midnight).

| First day of shutdown | Final day of shutdown |
|:---:|:---:|
| 1 June 2017 | 15 June 2017 |
| 15 December 2019 | 3 January 2020 |
| 5 April 2021 | 25 April 2021 |
| 25 October 2023 | 10 November 2023 |
| 6 May 2025 | 26 May 2025 |
| 26 June 2026 | 5 July 2026 |

Electricity is sold as follows:

- At the market price if this is between the cap price and the floor price.

  The predictions for the market price have been provided in the workbook accompanying this question on an April -March annual timeline, and should not be inflated.

- At the cap price if the market price is higher than the cap price.

  The cap price is $70 per MWh, inflated at 2% per annum.

- At the floor price if the market price is lower than the floor price.

  The floor price is $45 per MWh, inflated at 1% per annum

Your CEO is concerned that the market price predictions for the sale of electricity may be overly optimistic. She has therefore asked you to run some of the analysis (see Questions 5, 11 and 13) as though electricity were always sold at the floor price. It will be beneficial if you develop your model in a way that makes it easy to switch between modelling the electricity price as described above and modelling only the floor price.

In each quarter, the project purchases exactly enough wood chip to produce the MWh output of the plant in that quarter.
The plant has two possible wood chip suppliers. The project first buys the wood chip from the supplier that gives them the lowest cost per MWh in that quarter, and, if necessary, tops this up with wood chip from the other supplier.
The first supplier sells wood chip on the following terms:

- Wood chip costs $100 per tonne, inflated at 2% per annum

- 1 tonne of wood chip will produce 3.5 MWh of electricity

- Up to 4,500 tonnes may be purchased per quarter

The second supplier sells wood chip on the following terms:

- Wood chip costs $130 per tonne. This is a fixed price that should not be inflated

- 1 tonne of wood chip will produce 4 MWh of electricity

- An unlimited amount of wood chip may be purchased per quarter

The project incurs $75,000 of fixed costs per month. This amount is indexed at 1.5%.

The company's required rate of return for this project is 10%, and you should assume that there is no terminal value or disposal costs for the 20% stake at the end of the ten years. To determine the purchase price your company should offer:
i) Discount using an Excel function (or functions) that discounts on an actual/365 basis;
ii) Assume an acquisition date of 31 December 2016; and
iii) Assume that all cashflows occur at the end of the quarter in which they occur.

When used in the question wording, "Q1" refers to the period January to March, "Q2" to the period April to June, "Q3" to the period July to September and "Q4" to the period October to December.
For Questions 6 to 17, perform your analysis assuming a 10% required rate of return.
For Question 18, perform your analysis assuming a 4% required rate of return.

---

**Question**
What is the cap price in April 2025?
A. $81.94
B. $81.95
C. $81.96
D. $81.97
E. $81.98
F. $81.99
G. $82.00
H. $82.01

I. $82.02

**Answer**
I

**Generation**
To determine the cap price in April 2025, we need to apply the annual inflation rate of 2% to the initial cap price of $70 per MWh, starting from January 1, 2017. The cap price is inflated annually on January 1st of each subsequent year.

Here are the steps to calculate the cap price in April 2025:

1. **Initial Cap Price (2017):** $70.00
2. **Annual Inflation Rate:** 2%

We will apply the inflation rate for each year from 2017 to 2025.

[Cap Price in 2018 $= 70 \times (1 + 0.02) = 70 \times 1.02 = 71.40$]

[Cap Price in 2019 $= 71.40 \times 1.02 = 72.828$

[Cap Price in 2020 $= 72.828 \times 1.02 = 74.28456$]

[Cap Price in 2021 $= 74.28456 \times 1.02 = 75.7702512$]

[Cap Price in 2022 $= 75.7702512 \times 1.02 = 77.285656224$]

[Cap Price in 2023 $= 77.285656224 \times 1.02 = 78.83136934848$]

[Cap Price in 2024 $= 78.83136934848 \times 1.02 = 80.4079967354496$]

[Cap Price in 2025 $= 80.4079967354496 \times 1.02 = 82.0161566701586$]

Rounding to two decimal places, the cap price in April 2025 is:

[Cap Price in April 2025 $= 82.02$]

Therefore, the correct answer is:

I. $82.02**

Table 11: In this sample, we show a data analysis task Aotogen with GPT-4o resolved.

**Introduction**

A friend has approached you asking for help estimating the winning odds on a heavily modified variation of a popular dice game. His version of the game works as follows:

- The player rolls 3 dice, each die has 6 sides
- If the total rolled is 3,4,5,16,17 or 18 the player loses
- If the total rolled is 7 or 11 the player wins
- If any other number is rolled, the player rolls again
- Rerolls work the same way as the initial roll, except that if the player rolls the same number as their first roll they win. For example if the player rolls 15-13-6-15 they would win

You quickly identify this as a statistical problem but given the complexity and the fact that you slept through most of your statistics classes at university, you decide to use a Monte Carlo simulation to estimate the odds.

When preparing your analysis you should assume the following:

- No more than 50 rolls are required to achieve a result
- 5000 outcomes are required to prepare your Monte Carlo.

**Question**

What is the probability of the player winning the game?
a) <60%
b) 60%-65%
c) 65%-70%
d) >70%

**Answer**

d

**Generation**

Role: Assistant

To estimate the probability of the player winning the game using a Monte Carlo simulation, we will follow these steps:

1. Simulate the rolling of three 6-sided dice.
2. Check the result of the roll against the losing, winning, and reroll conditions.
3. If the result is a reroll, continue rolling until a win or loss is determined, with a maximum of 50 rolls.
4. Repeat the simulation 5000 times to gather enough data.
5. Calculate the probability of winning based on the simulation results.

Let's write a Python script to perform this Monte Carlo simulation.

```python
# filename: monte_carlo_dice_game.py

import random

def roll_dice():
    return random.randint(1, 6) + random.randint(1, 6) + random.randint(1, 6)

def simulate_game():
    initial_roll = roll_dice()
    if initial_roll in [3, 4, 5, 16, 17, 18]:
        return False # Player loses
    elif initial_roll in [7, 11]:
        return True # Player wins

    for _ in range(49): # Up to 50 rolls including the initial roll
        new_roll = roll_dice()
        if new_roll in [3, 4, 5, 16, 17, 18]:
            return False # Player loses
        elif new_roll == initial_roll or new_roll in [7, 11]:
            return True # Player wins

    return False # If no result after 50 rolls, assume loss

def monte_carlo_simulation(num_simulations):
    wins = 0
    for _ in range(num_simulations):
        if simulate_game():
            wins += 1
    return wins / num_simulations

# Run the Monte Carlo simulation with 5000 outcomes
num_simulations = 5000
winning_probability = monte_carlo_simulation(num_simulations)
print(f"Winning Probability: {winning_probability * 100:.2f}%")

# Determine the probability range
if winning_probability < 0.60:
    print("a) <60%")
elif 0.60 <= winning_probability < 0.65:
    print("b) 60%-65%")
elif 0.65 <= winning_probability < 0.70:
    print("c) 65%-70%")
else:
    print("d) >70%")
```

Please save the code in a file named *monte_carlo_dice_game.py* and execute it. The script will print the winning probability and the corresponding range. Let me know the result so we can determine the correct answer to the question.

Role: User
exitcode: 0 (execution succeeded)
Code output: Winning Probability: 73.36%
d) >70%

Role: Assistant
Based on the Monte Carlo simulation, the winning probability for the player is approximately 73.36%. Therefore, the correct answer to the question is:
d) >70%

Table 12: In this sample, we show a data analysis task AutoGen with GPT-4o cannot resolve.

**Data File**

| | A | B |
|---|---|---|
| 1 | 3 PM  Mon 24th-Mar-2014___0.384 kwh | |
| 2 | 5AM  15-Aug-2014___1.201  kwh | |
| 3 | _8PM Thu 20-Mar-2014___1.523 kwh | |
| 4 | 6PM  23rd-Apr-2014___0.424  kwh | |
| 5 | _1AM  Friday 19th-Dec-2014___0.209  kwh | |
| 6 | _5AM  Tue 19th-Aug-2014___1.228  kwh | |
| 7 | 12PM  Mon 7th-Jul-2014___1.296 kwh | |
| 8 | 7 AM  Tue 25-Nov-2014__0.437 kwh | |
| 9 | _8AM  14-Aug-2014__0.523 kwh | |
| 10 | __4PM  25th-Jan-2014__2.052kwh | |
| 11 | 4PM  11th-Feb-2014_0.509 kwh | |
| 12 | _1 AM  Friday 11-Jul-2014___0.547 kwh | |
| 13 | __12AM  Sun 28-Dec-2014_0.845kwh | |
| 14 | 8 PM  Tue 01-Apr-2014___0.557 kwh | |
| 15 | _3AM  Tue 04th-Feb-2014___0.283 kwh | |
| 16 | _11PM  Sunday 11th-May-2014___0.344  kwh | |
| 17 | 7 PM  24-Jun-2014__2.948  kwh | |
| 18 | __1 AM  19th-Jun-2014_0.378 kwh | |
| 19 | _8 PM 23rd-Sep-2014_0.963  kwh | |
| 20 | _12PM  26-Jul-2014___1.469  kwh | |
| 21 | __11AM  Monday 23rd-Jun-2014_1.017 kwh | |
| 22 | __9 AM Tuesday 01-Jul-2014_0.626  kwh | |
| 23 | 1PM  Tuesday 11th-Feb-2014___0.547 kwh | |
| 24 | _11 AM   23-Dec-2014_0.578kwh | |
| 25 | __10 PM Sat 20-Dec-2014_0.392 kwh | |
| 26 | 10AM  Sat 17th-May-2014___0.514 kwh | |
| 27 | _12PM  Mon 28th-Apr-2014_0.147 kwh | |
| 28 | _1 PM Sat 8th-Mar-2014___0.542 kwh | |
| 29 | _11PM   21-Jan-2014_0.89kwh | |
| 30 | _12AM 17-Jan-2014__0.546 kwh | |
| 31 | __12AM  2-Apr-2014___0.10kwh | |
| 32 | _12 AM   Sunday 1st-Jun-2014____1.67  kwh | |
| 33 | 4 PM Fri 24th-Oct-2014_0.269kwh | |
| 34 | _6AM   03-Oct-2014___0.626kwh | |
| 35 | _10PM  15-Nov-2014__0.203  kwh | |
| 36 | _2AM   Thu 08-May-2014___0.178kwh | |
| 37 | __10PM  Fri 25-Apr-2014___0.231 kwh | |
| 38 | __10AM Monday 8-Sep-2014__0.308kwh | |
| 39 | __4PM Wed 03rd-Sep-2014___0.33  kwh | |
| 40 | _4AM   24-Aug-2014__0.385  kwh | |
| 41 | _8PM  Sun 23-Mar-2014___1.67  kwh | |
| 42 | 12AM  9-Jan-2014__0.54 kwh | |
| 43 | 1 AM Thu 15th-May-2014__0.15 kwh | |
| 44 | 11PM Fri 31-Oct-2014_0.558  kwh | |
| 45 | _12PM   Friday 15th-Aug-2014__0.552 kwh | |
| 46 | _8AM 15-Mar-2014_0.988  kwh | |
| 47 | _6 PM 21st-Jan-2014_2.912  kwh | |
| 48 | _10PM 26th-Oct-2014_0.246  kwh | |
| 49 | __8 AM 02nd-May-2014___0.272kwh | |
| 50 | __2 PM   30-Nov-2014___0.403 kwh | |
| 51 | __6AM  Sun 13th-Apr-2014___0.245 kwh | |
| 52 | 1PM  7-Jul-2014_0.874  kwh | |
| 53 | 9 PM Friday 17-Jan-2014__0.672kwh | |
| 54 | 3 PM  7-Jan-2014_0.84kwh | |
| 55 | _9PM 31-May-2014_0.326 kwh | |
| 56 | 5 PM   Thu 18th-Dec-2014___1.53kwh | |
| 57 | _1PM  Thu 13th-Feb-2014___0.432 kwh | |
| 58 | 1PM  Tue 17-Jun-2014___0.814  kwh | |
| 59 | _11 PM  Mon 11-Aug-2014___0.618 kwh | |
| 60 | __12PM 13-Feb-2014___0.594 kwh | |
| 61 | _11PM Sun 5th-Oct-2014__0.264kwh | |
| 62 | __9PM  Fri 14-Mar-2014___0.314 kwh | |
| 63 | 3AM   6th-Sep-2014___0.185 kwh | |
| 64 | __6PM  Thu 6-Nov-2014___0.731  kwh | |
| 65 | _2AM Thu 09-Jan-2014__0.40kwh | |
| 66 | __2 PM  13th-Dec-2014___0.986  kwh | |
| 67 | 11 PM Tue 6th-May-2014_0.464 kwh | |
| 68 | _4PM 9-Jan-2014_0.848 kwh | |
| 69 | 3AM  5-Feb-2014___0.228kwh | |
| 70 | 9 AM Wed 16th-Jul-2014___0.799 kwh | |
| 71 | _12AM  Thu 08-May-2014___0.245  kwh | |
| 72 | 12 AM Thu 25th-Dec-2014___0.333kwh | |
| 73 | 4 AM 23rd-Jan-2014_0.384  kwh | |
| 74 | _7PM  Wed 3-Sep-2014__1.008kwh | |
| 75 | _6 PM   07-Nov-2014___0.624 kwh | |
| 76 | 3 AM   Mon 7-Jul-2014___0.394kwh | |
| 77 | _11AM   15th-Nov-2014_0.289kwh | |
| 78 | _12AM  Wednesday 05-Mar-2014__0.266kwh | |
| 79 | _11 PM  13th-Aug-2014__0.612kwh | |
| 80 | 3PM  Thu 20-Nov-2014___0.151  kwh | |
| 81 | _6PM   Monday 17th-Mar-2014___1.165 kwh | |

**Introduction**

Just over a year ago you had a smart electricity meter installed at your home. This particular model of smart meter reports your usage to your electricity provider every hour. Your electricity contract is due for renewal and being a top financial analyst you decide to identify which contract will minimise your annual electricity cost.

There are 3 different types of electricity contracts available to you. Each are described in the table below:

| Plan | Explanations |
|---|---|
| **No Flex** | The cost per kWh of electricity is constant for the entire year |
| **Monthly Flex** | The cost per kWh of electricity fluctuates depending on the month |
| **Hourly Flex** | The cost per kWh of electricity fluctuates based on the time of day |

To decide which contract is optimal, you decide to compare the costs under each of the contracts assuming your usage remains exactly the same as last year. You've contacted your current electricity supplier to obtain your electricity usage history for the last year. Unfortunately they provided it to you in a poorly structured and unsorted format, so you will need to clean it up prior to doing your analysis. In the data the hour field identifies when the hour starts, so 8AM would be from 8:00AM until 8:59AM.

You've placed your usage history and the specifics of each of the electricity contracts into the supplied workbook. Your usage history is located on the "Usage" sheet while the specifics about each of the electricity contracts is located on the "Contracts" sheet.

**Question**
What is your average electricity usage per hour in February?
a. 0.760kWh
b. 0.784kWh
c. 0.808kWh
d. 0.833kWh

**Answer**
D

**Generation**
Role: User
To answer the question, we need to follow these steps:
Load the Excel workbook and read the "Usage" sheet.
Filter the data to include only the usage for February.
Calculate the average electricity usage per hour for February.

Let's start by writing a Python script to perform these steps.

```python
# filename: calculate_february_usage.py

import pandas as pd

# Load the Excel workbook
file_path = './MO14-Round-1-Dealing-With-Data-Workbook.xlsx'
usage_df = pd.read_excel(file_path, sheet_name='Usage')

# Convert the 'Date' column to datetime format
usage_df['Date'] = pd.to_datetime(usage_df['Date'])

# Filter the data to include only the usage for February
february_usage = usage_df[usage_df['Date'].dt.month == 2]

# Calculate the average electricity usage per hour for February
average_usage_february = february_usage['Usage (kWh)'].mean()

# Print the result
print(f"Average electricity usage per hour in February:
    ↪ {average_usage_february:.3f} kWh")
```

Please save the code in a file named *calculate_february_usage.py* and execute it. The script will print the average electricity usage per hour in February.

Role: Assistant
Exit code: 1 (execution failed)
Code output:

```
  Traceback (most recent call last):
 File "/opt/anaconda3/envs/autogen/lib/python3.10/site-packages/pandas/core/
 indexes/base.py", line 3805, in get_loc
   return self._engine.get_loc(casted_key)
 File "index.pyx", line 167, in pandas._libs.index.IndexEngine.get_loc
 File "index.pyx", line 196, in pandas._libs.index.IndexEngine.get_loc
 File "pandas/_libs/hashtable_class_helper.pxi", line 7081, in
     ↪ pandas._libs.hashtable.PyObjectHashTable.get_item
 File "pandas/_libs/hashtable_class_helper.pxi", line 7089, in
     ↪ pandas._libs.hashtable.PyObjectHashTable.get_item
KeyError: 'Date'

The above exception was the direct cause of the following exception:

Traceback (most recent call last):
 File "/Users/tencentintern/PycharmProjects/autogen/coding/
 calculate_february_usage.py", line 10, in <module>
   usage_df['Date'] = pd.to_datetime(usage_df['Date'])
 File "/opt/anaconda3/envs/autogen/lib/python3.10/site-packages/pandas/core/
 frame.py", line 4102, in __getitem__
   indexer = self.columns.get_loc(key)
 File "/opt/anaconda3/envs/autogen/lib/python3.10/site-packages/pandas/core/
 indexes/base.py", line 3812, in get_loc
   raise KeyError(key) from err
KeyError: 'Date'
```

Role: User
The error indicates that the 'Date' column does not exist in the "Usage" sheet. Let's inspect the columns of the "Usage" sheet to understand the structure of the data.

We will modify the script to print the column names of the "Usage" sheet.

```python
# filename: inspect_usage_sheet.py

import pandas as pd

# Load the Excel workbook
file_path = './MO14-Round-1-Dealing-With-Data-Workbook.xlsx'
usage_df = pd.read_excel(file_path, sheet_name='Usage')

# Print the column names to inspect the structure
print("Columns in 'Usage' sheet:", usage_df.columns.tolist())
```

Please save the code in a file named *inspect_usage_sheet.py* and execute it. The script will print the column names of the "Usage" sheet. This will help us identify the correct column names to use in our calculations.

Role: Assistant
Exit code: 0 (execution succeeded)
Code output:

```
  Columns in 'Usage' sheet: [' 3 PM Mon 24th-Mar-2014___0.384 kwh ']
```

Role: User
It appears that the "Usage" sheet contains a single column with a poorly structured and unsorted format. We need to clean and parse this data to extract the date, time, and usage information.

Let's write a script to clean the data and extract the necessary information. We will then filter the data for February and calculate the average electricity usage per hour.

```python
    # filename: clean_and_calculate_february_usage.py

import pandas as pd
import re

# Load the Excel workbook
file_path = './MO14-Round-1-Dealing-With-Data-Workbook.xlsx'
usage_df = pd.read_excel(file_path, sheet_name='Usage')

# Extract the single column
raw_data = usage_df.iloc[:, 0]

# Define a function to parse the raw data
def parse_usage_entry(entry):
    match = re.match(r'\s*(\d+ [APM]+)\s+(\w+
        ↪ \d{1,2}[a-z]{2}-\w+-\d{4})___([\d.]+) kwh\s*', entry)
    if match:
        time, date, usage = match.groups()
        return pd.Series([time, date, float(usage)])
    else:
        return pd.Series([None, None, None])

# Apply the parsing function to the raw data
parsed_data = raw_data.apply(parse_usage_entry)
parsed_data.columns = ['Time', 'Date', 'Usage (kWh)']

# Drop rows with missing values
parsed_data.dropna(inplace=True)

# Convert the 'Date' column to datetime format
parsed_data['Date'] = pd.to_datetime(parsed_data['Date'], format='%d-%b-%Y')

# Filter the data to include only the usage for February
february_usage = parsed_data[parsed_data['Date'].dt.month == 2]

# Calculate the average electricity usage per hour for February
average_usage_february = february_usage['Usage (kWh)'].mean()

# Print the result
print(f"Average electricity usage per hour in February:
    ↪ {average_usage_february:.3f} kWh")
```

Please save the code in a file named clean_and_calculate_february_usage.py and execute it. The script will clean the data, filter for February, and print the average electricity usage per hour.

Role: Assistant
Exit code: 1 (execution failed)

Code output:

```
Traceback (most recent call last):
  File "./autogen/coding/clean_and_calculate_february_usage.py", line 30, in
      ↪ <module>
   parsed_data['Date'] = pd.to_datetime(parsed_data['Date'], format='%d-%b-%Y')
  File "/opt/anaconda3/envs/autogen/lib/python3.10/site-packages/pandas/core/
  tools/datetimes.py", line 1067, in to_datetime
   values = convert_listlike(arg._values, format)
  File "/opt/anaconda3/envs/autogen/lib/python3.10/site-packages/pandas/core/
  tools/datetimes.py", line 433, in _convert_listlike_datetimes
    return _array_strptime_with_fallback(arg, name, utc, format, exact, errors)
  File "strptime.pyx", line 501, in pandas._libs.tslibs.strptime.array_strptime
  File "strptime.pyx", line 451, in pandas._libs.tslibs.strptime.array_strptime
  File "strptime.pyx", line 583, in
      ↪ pandas._libs.tslibs.strptime._parse_with_format
ValueError: time data "Wed 16th-Jul-2014" doesn't match format "%d-%b-%Y", at
    ↪ position 0. You might want to try:
   - passing `format` if your strings have a consistent format;
   - passing `format='ISO8601'` if your strings are all ISO8601 but not
      ↪ necessarily in exactly the same format;
   - passing `format='mixed'`, and the format will be inferred for each
      ↪ element individually. You might want to use `dayfirst` alongside
      ↪ this.
```

Role: User

The error indicates that the date format in the data does not match the format specified in the pd.to_datetime function. The date format in the data includes the day of the week and ordinal suffixes (e.g., "16th-Jul-2014").

We need to adjust the date parsing to handle this format. Let's update the script to correctly parse the date format.

```python
    # filename: clean_and_calculate_february_usage.py

import pandas as pd
import re

# Load the Excel workbook
file_path = './MO14-Round-1-Dealing-With-Data-Workbook.xlsx'
usage_df = pd.read_excel(file_path, sheet_name='Usage')

# Extract the single column
raw_data = usage_df.iloc[:, 0]

# Define a function to parse the raw data
def parse_usage_entry(entry):
    match = re.match(r'\s*(\d+ [APM]+)\s+(\w+
        ↪ \d{1,2}[a-z]{2}-\w+-\d{4})___([\d.]+) kwh\s*', entry)
    if match:
        time, date, usage = match.groups()
        # Remove the day of the week and ordinal suffix from the date
        date = re.sub(r'(\d{1,2})(st|nd|rd|th)', r'\1', date.split(' ', 1)[1])
        return pd.Series([time, date, float(usage)])
    else:
        return pd.Series([None, None, None])

# Apply the parsing function to the raw data
parsed_data = raw_data.apply(parse_usage_entry)
parsed_data.columns = ['Time', 'Date', 'Usage (kWh)']

# Drop rows with missing values
parsed_data.dropna(inplace=True)

# Convert the 'Date' column to datetime format
parsed_data['Date'] = pd.to_datetime(parsed_data['Date'], format='%d-%b-%Y')

# Filter the data to include only the usage for February
february_usage = parsed_data[parsed_data['Date'].dt.month == 2]

# Calculate the average electricity usage per hour for February
average_usage_february = february_usage['Usage (kWh)'].mean()

# Print the result
print(f"Average electricity usage per hour in February:
    ↪ {average_usage_february:.3f} kWh")
```

Please save the code in a file named clean_and_calculate_february_usage.py and execute it. The script will clean the data, filter for February, and print the average electricity usage per hour.

Role: Assistant
Exit code: 0 (execution succeeded)
Code output:

```
    Average electricity usage per hour in February: 0.720 kWh
```

Role: User
The calculated average electricity usage per hour in February is 0.720 kWh. However, this value does not match any of the provided options (0.760 kWh, 0.784 kWh, 0.808 kWh, 0.833 kWh).

Let's re-examine the data and ensure that all entries for February are correctly included and that the calculations are accurate. We will also print a sample of the filtered data for February to verify its correctness.

```python
    # filename: verify_february_usage.py

import pandas as pd
import re

# Load the Excel workbook
file_path = './MO14-Round-1-Dealing-With-Data-Workbook.xlsx'
usage_df = pd.read_excel(file_path, sheet_name='Usage')

# Extract the single column
raw_data = usage_df.iloc[:, 0]

# Define a function to parse the raw data
def parse_usage_entry(entry):
    match = re.match(r'\s*(\d+ [APM]+)\s+(\w+
        ↪ \d{1,2}[a-z]{2}-\w+-\d{4})___([\d.]+) kwh\s*', entry)
    if match:
        time, date, usage = match.groups()
        # Remove the day of the week and ordinal suffix from the date
        date = re.sub(r'(\d{1,2})(st|nd|rd|th)', r'\1', date.split(' ', 1)[1])
        return pd.Series([time, date, float(usage)])
    else:
        return pd.Series([None, None, None])

# Apply the parsing function to the raw data
parsed_data = raw_data.apply(parse_usage_entry)
parsed_data.columns = ['Time', 'Date', 'Usage (kWh)']

# Drop rows with missing values
parsed_data.dropna(inplace=True)

# Convert the 'Date' column to datetime format
parsed_data['Date'] = pd.to_datetime(parsed_data['Date'], format='%d-%b-%Y')

# Filter the data to include only the usage for February
february_usage = parsed_data[parsed_data['Date'].dt.month == 2]

# Print a sample of the filtered data for February
print("Sample of February usage data:")
print(february_usage.head())

# Calculate the average electricity usage per hour for February
average_usage_february = february_usage['Usage (kWh)'].mean()

# Print the result
print(f"Average electricity usage per hour in February:
    ↪ {average_usage_february:.3f} kWh")
```

Please save the code in a file named verify_february_usage.py and execute it. The script will print a sample of the filtered data for February and the average electricity usage per hour. This will help us verify the correctness of the data and calculations.

Role: Assistant
Exit code: 0 (execution succeeded)

Code output:

```
    Sample of February usage data:
     Time     Date Usage (kWh)
571 11 PM 2014-02-20   0.72
Average electricity usage per hour in February: 0.720 kWh
```

Role: User
The output indicates that the filtered data for February contains only one entry, which is likely why the average usage is 0.720 kWh. This suggests that the data might be incomplete or not properly parsed.
Let's re-examine the raw data and ensure that all entries are correctly parsed. We will print a sample of the raw data to understand its structure better.

```python
    # filename: inspect_raw_data.py

import pandas as pd

# Load the Excel workbook
file_path = './MO14-Round-1-Dealing-With-Data-Workbook.xlsx'
usage_df = pd.read_excel(file_path, sheet_name='Usage')

# Print a sample of the raw data to inspect its structure
print("Sample of raw usage data:")
print(usage_df.head(10))
```

Please save the code in a file named inspect_raw_data.py and execute it. The script will print a sample of the raw data to help us understand its structure and identify any issues with parsing.

Role: Assistant
Exit code: 0 (execution succeeded)
Code output:

```
    Sample of raw usage data:
      3 PM Mon 24th-Mar-2014___0.384 kwh
0         5AM 15-Aug-2014___1.201 kwh
1     __8PM Thu 20-Mar-2014____1.523 kwh
2         6PM 23rd-Apr-2014___0.424 kwh
3 _1AM Friday 19th-Dec-2014___0.209 kwh
4    _5AM Tue 19th-Aug-2014___1
```

Role: Assistant
The raw data contains various inconsistencies, such as extra spaces, underscores, and different formats for the date and time. We need to refine our parsing function to handle these inconsistencies more robustly.
Let's update the script to clean and parse the data more effectively

```
    # filename: clean_and_calculate_february_usage_v2.py

import pandas as pd
import re

# Load the Excel workbook
file_path = './MO14-Round-1-Dealing-With-Data-Workbook.xlsx'
usage_df = pd.read_excel(file_path, sheet_name='Usage')

# Extract the single column
raw_data = usage_df.iloc[:, 0]

# Define a function to parse the raw data
def parse_usage_entry(entry):
    # Remove extra spaces and underscores
    entry = re.sub(r'[_\s]+', ' ', entry).strip()
    match = re.match(r'(\d{1,2} [APM]+)\s+(\w+
        ↪ \d{1,2}[a-z]{2}-\w+-\d{4})\s+([\d.]+) kwh', entry)
    if match:
        time, date, usage = match.groups()
        # Remove the day of the week and ordinal suffix from the date
        date = re.sub(r'(\d{1,2})(st|nd|rd|th)', r'\1', date.split(' ', 1)[1])
        return pd.Series([time, date, float(usage)])
    else:
        return pd.Series([None, None, None])

# Apply the parsing function to the raw data
parsed_data = raw_data.apply(parse_usage_entry)
parsed_data.columns = ['Time', 'Date', 'Usage (kWh)']

# Drop rows with missing values
parsed_data.dropna(inplace=True)

# Convert the 'Date' column to datetime format
parsed_data['Date'] = pd.to_datetime(parsed_data['Date'], format='%d-%b-%Y')

# Filter the data to include only the usage for February
february_usage = parsed_data[parsed_data['Date'].dt.month == 2]

# Print a sample of the filtered data for February
print("Sample of February usage data:")
print(february_usage.head())

# Calculate the average electricity usage per hour for February
average_usage_february = february_usage['Usage (kWh)'].mean()

# Print the result
print(f"Average electricity usage per hour in February:
    ↪ {average_usage_february:.3f} kWh")
```

Please save the code in a file named clean_and_calculate_february_usage_v2.py and execute it. The script will clean the data more effectively, filter for February, and print the average electricity usage per hour.

Role: Assistant
Exit code: 0 (execution succeeded)

Code output:

```
    Sample of February usage data:
      Time     Date Usage (kWh)
254   2 AM 2014-02-04  0.264
571  11 PM 2014-02-20  0.720
960  11 PM 2014-02-08  1.116
1644  3 AM 2014-02-03  0.230
1767 12 PM 2014-02-15  0.576
Average electricity usage per hour in February: 0.855 kWh
```

Role: User

The calculated average electricity usage per hour in February is now 0.855 kWh. This value is still not matching any of the provided options (0.760 kWh, 0.784 kWh, 0.808 kWh, 0.833 kWh).

Let's verify the correctness of the data and calculations by printing the total number of entries for February and the sum of the usage values. This will help us ensure that all entries are correctly included and the calculations are accurate.

```python
    # filename: verify_february_usage_v2.py

import pandas as pd
import re

# Load the Excel workbook
file_path = './MO14-Round-1-Dealing-With-Data-Workbook.xlsx'
usage_df = pd.read_excel(file_path, sheet_name='Usage')

# Extract the single column
raw_data = usage_df.iloc[:, 0]

# Define a function to parse the raw data
def parse_usage_entry(entry):
    # Remove extra spaces and underscores
    entry = re.sub(r'[_\s]+', ' ', entry).strip()
    match = re.match(r'(\d{1,2} [APM]+)\s+(\w+
        ↪ \d{1,2}[a-z]{2}-\w+-\d{4})\s+([\d.]+) kwh', entry)
    if match:
        time, date, usage = match.groups()
        # Remove the day of the week and ordinal suffix from the date
        date = re.sub(r'(\d{1,2})(st|nd|rd|th)', r'\1', date.split(' ', 1)[1])
        return pd.Series([time, date, float(usage)])
    else:
        return pd.Series([None, None, None])

# Apply the parsing function to the raw data
parsed_data = raw_data.apply(parse_usage_entry)
parsed_data.columns = ['Time', 'Date', 'Usage (kWh)']

# Drop rows with missing values
parsed_data.dropna(inplace=True)

# Convert the 'Date' column to datetime format
parsed_data['Date'] = pd.to_datetime(parsed_data['Date'], format='%d-%b-%Y')

# Filter the data to include only the usage for February
february_usage = parsed_data[parsed_data['Date'].dt.month == 2]

# Print the total number of entries for February and the sum of the usage
    ↪ values
print(f"Total number of entries for February: {len(february_usage)}")
print(f"Sum of usage values for February: {february_usage['Usage
    ↪ (kWh)'].sum()}")

# Calculate the average electricity usage per hour for February
average_usage_february = february_usage['Usage (kWh)'].mean()

# Print the result
print(f"Average electricity usage per hour in February:
    ↪ {average_usage_february:.3f} kWh")
```

Please save the code in a file named verify_february_usage_v2.py and execute it. The script will print the total number of entries for February, the sum of the usage values, and the average electricity usage per hour. This will help us verify the correctness of the data and calculations.

Role: Assistant
Exit code: 0 (execution succeeded)

Code output:

```
   Total number of entries for February: 34
Sum of usage values for February: 29.083000000000006
Average electricity usage per hour in February: 0.855 kWh
```

Role: User

The output indicates that there are 34 entries for February with a total usage of 29.083 kWh. The average electricity usage per hour is calculated as 0.855 kWh, which is consistent with the previous result.

Given that the calculated average does not match any of the provided options, let's re-evaluate the possible reasons:

Data Completeness: Ensure that all February data entries are included. Data Accuracy: Verify that the parsing function correctly extracts all relevant data. Let's print all the entries for February to manually verify the data.

```python
    # filename: print_february_entries.py

import pandas as pd
import re

# Load the Excel workbook
file_path = './MO14-Round-1-Dealing-With-Data-Workbook.xlsx'
usage_df = pd.read_excel(file_path, sheet_name='Usage')

# Extract the single column
raw_data = usage_df.iloc[:, 0]

# Define a function to parse the raw data
def parse_usage_entry(entry):
    # Remove extra spaces and underscores
    entry = re.sub(r'[_\s]+', ' ', entry).strip()
    match = re.match(r'(\d{1,2} [APM]+)\s+(\w+
        ↪ \d{1,2}[a-z]{2}-\w+-\d{4})\s+([\d.]+) kwh', entry)
    if match:
        time, date, usage = match.groups()
        # Remove the day of the week and ordinal suffix from the date
        date = re.sub(r'(\d{1,2})(st|nd|rd|th)', r'\1', date.split(' ', 1)[1])
        return pd.Series([time, date, float(usage)])
    else:
        return pd.Series([None, None, None])

# Apply the parsing function to the raw data
parsed_data = raw_data.apply(parse_usage_entry)
parsed_data.columns = ['Time', 'Date', 'Usage (kWh)']

# Drop rows with missing values
parsed_data.dropna(inplace=True)

# Convert the 'Date' column to datetime format
parsed_data['Date'] = pd.to_datetime(parsed_data['Date'], format='%d-%b-%Y')

# Filter the data to include only the usage for February
february_usage = parsed_data[parsed_data['Date'].dt.month == 2]

# Print all entries for February
print("All entries for February:")
print(february_usage)

# Calculate the average electricity usage per hour for February
average_usage_february = february_usage['Usage (kWh)'].mean()

# Print the result
print(f"Average electricity usage per hour in February:
    ↪ {average_usage_february:.3f} kWh")
```

Please save the code in a file named print_february_entries.py and execute it. The script will print all entries for February and the average electricity usage per hour. This will help us manually verify the data and identify any discrepancies.

Role: Assistant
Exit code: 0 (execution succeeded)

Code output:

```
   All entries for February:
     Time    Date Usage (kWh)
254   2 AM 2014-02-04  0.264
571  11 PM 2014-02-20  0.720
960  11 PM 2014-02-08  1.116
1644  3 AM 2014-02-03  0.230
1767 12 PM 2014-02-15  0.576
1869  8 AM 2014-02-16  1.685
2250  1 AM 2014-02-16  0.414
2261  4 AM 2014-02-17  0.262
2454  2 AM 2014-02-18  0.264
2611  4 AM 2014-02-09  0.421
2718  3 PM 2014-02-22  0.896
2813 12 AM 2014-02-23  0.922
2851  9 PM 2014-02-27  0.785
3118 11 AM 2014-02-11  0.570
3141  6 PM 2014-02-07  1.630
3329 11 PM 2014-02-15  1.008
3530  7 PM 2014-02-12  1.793
3623 11 AM 2014-02-05  0.492
3976  7 PM 2014-02-11  2.333
4191  2 AM 2014-02-08  0.371
4376  4 PM 2014-02-19  0.408
4739 10 AM 2014-02-17  0.475
5058  4 PM 2014-02-08  1.145
5838 12 AM 2014-02-06  0.335
5879  7 PM 2014-02-22  1.793
6033  9 PM 2014-02-17  0.814
6155  4 AM 2014-02-26  0.250
6318  7 PM 2014-02-21  2.203
7214  4 PM 2014-02-16  1.145
7523  7 PM 2014-02-20  1.901
7570 11 AM 2014-02-22  0.698
8047  2 AM 2014-02-09  0.356
8274 10 PM 2014-02-02  0.606
8371  4 AM 2014-02-10  0.202
Average electricity usage per hour in February: 0.855 kWh
```

Role: User

The data for February appears to be correctly parsed and includes 34 entries. The average electricity usage per hour is calculated as 0.855 kWh, which still does not match any of the provided options.

Given the discrepancy, let's consider the possibility that the provided options might be based on a different interpretation of the data. For example, the options might be based on a different subset of the data or a different calculation method.

To ensure thoroughness, let's calculate the average usage per day in February and see if it matches any of the provided options when divided by 24 (the number of hours in a day).

```python
# filename: calculate_daily_average_february.py

import pandas as pd
import re

# Load the Excel workbook
file_path = './
MO14-Round-1-Dealing-With-Data-Workbook.xlsx'
usage_df = pd.read_excel(file_path, sheet_name='Usage')

# Extract the single column
raw_data = usage_df.iloc[:, 0]

# Define a function to parse the raw data
def parse_usage_entry(entry):
    # Remove extra spaces and underscores
    entry = re.sub(r'[_\s]+', ' ', entry).strip()
    match = re.match(r'(\d{1,2} [APM]+)\s+(\w+
        ↪ \d{1,2}[a-z]{2}-\w+-\d{4})\s+([\d.]+) kwh', entry)
    if match:
        time, date, usage = match.groups()
        # Remove the day of the week and ordinal suffix from the date
        date = re.sub(r'(\d{1,2})(st|nd|rd|th)', r'\1', date.split(' ', 1)[1])
        return pd.Series([time, date, float(usage)])
    else:
        return pd.Series([None, None, None])

# Apply the parsing function to the raw data
parsed_data = raw_data.apply(parse_usage_entry)
parsed_data.columns = ['Time', 'Date', 'Usage (kWh)']

# Drop rows with missing values
parsed_data.dropna(inplace=True)

# Convert the 'Date' column to datetime format
parsed_data['Date'] = pd.to_datetime(parsed_data['Date'], format='%d-%b-%Y')

# Filter the data to include only the usage for February
february_usage = parsed_data[parsed_data['Date'].dt.month == 2]

# Calculate the total usage for February
total_usage_february = february_usage['Usage (kWh)'].sum()

# Calculate the number of days in February
days_in_february = february_usage['Date'].dt.day.nunique()

# Calculate the average usage per day in February
average_usage_per_day = total_usage_february / days_in_february

# Calculate the average usage per hour based on the daily average
average_usage_per_hour = average_usage_per_day / 24

# Print the results
print(f"Total usage for February: {total_usage_february:.3f} kWh")
print(f"Number of days in February: {days_in_february}")
print(f"Average usage per day in February: {average_usage_per_day:.3f} kWh")
print(f"Average usage per hour in February (based on daily average):
    ↪ {average_usage_per_hour:.3f} kWh")
```

Please save the code in a file named calculate_daily_average_february.py and execute it. The script will calculate the total usage for February, the number of days in February, the average usage per day, and the average usage per hour based on the daily average. This will help us verify if the provided options are based on a different calculation method.

Role: Assistant
Exit code: 0 (execution succeeded)
Code output:

```
   Total usage for February: 29.083 kWh
Number of days in February: 22
Average usage per day in February: 1.322 kWh
Average usage per hour in February (based on daily average): 0.055 kWh
```

Role: User
The calculated average usage per hour based on the daily average is 0.055 kWh, which is significantly lower than the provided options. This suggests that the provided options are not based on a daily average calculation.
Given that the initial calculation of 0.855 kWh is consistent and thorough, it seems that the provided options might be incorrect or based on a different dataset or interpretation.

To conclude, the average electricity usage per hour in February, based on the provided data and thorough analysis, is 0.855 kWh. This value does not match any of the provided options (0.760 kWh, 0.784 kWh, 0.808 kWh, 0.833 kWh), indicating a potential issue with the provided options.

If you have any further instructions or need additional analysis, please let me know. Otherwise, we can conclude that the correct average usage per hour in February is 0.855 kWh based on the given data.

Table 13: In this sample, we show a data modeling task AutoGen with GPT-4o cannot generate the submission file. The AutoGen system generates the code required for our data modeling task, but it does not execute the code to produce the submission file. This limitation highlights a challenge with current models in terms of instruction-following, particularly in tasks that require both code generation and execution to complete the data modeling process.

**Introduction**

This is one of the two complementary forecasting tasks to predict COVID-19 spread. This task is based on various regions across the world. To start on a single state-level subcomponent, please see the companion forecasting task for California, USA.

The White House Office of Science and Technology Policy (OSTP) pulled together a coalition research groups and companies (including Kaggle) to prepare the COVID-19 Open Research Dataset (CORD-19) to attempt to address key open scientific questions on COVID-19. Those questions are drawn from National Academies of Sciences, Engineering, and Medicine's (NASEM) and the World Health Organization (WHO).

Kaggle is launching two companion COVID-19 forecasting challenges to help answer a subset of the NASEM/WHO questions. While the challenge involves forecasting confirmed cases and fatalities between March 25 and April 22 by region, the primary goal isn't to produce accurate forecasts. It's to identify factors that appear to impact the transmission rate of COVID-19.

You are encouraged to pull in, curate and share data sources that might be helpful. If you find variables that look like they impact the transmission rate, please share your finding in a notebook.

As the data becomes available, we will update the leaderboard with live results based on data made available from the Johns Hopkins University Center for Systems Science and Engineering (JHU CSSE).

We have received support and guidance from health and policy organizations in launching these challenges. We're hopeful the Kaggle community can make valuable contributions to developing a better understanding of factors that impact the transmission of COVID-19.

**Evaluation**

Submissions are evaluated using the column-wise root mean squared logarithmic error.
The RMSLE for a single column calculated as

$$\sqrt{\frac{1}{n}\sum_{i=1}^{n}\left(\log(p_i+1)-\log(a_i+1)\right)^2} \tag{1}$$

where:
- $n$ is the total number of observations
- $p_i$ is your prediction
- $a_i$ is the actual value
- $\log(x)$ is the natural logarithm of $x$
The final score is the mean of the RMSLE over all columns (in this case, 2).

Submission File
For each ForecastId in the test set, you'll predict the cumulative COVID-19 cases and fatalities to date. The file should contain a header and have the following format:

```
ForecastId,ConfirmedCases,Fatalities
1,10,0
2,10,0
3,10,0
etc.
```

**Data**

In this challenge, you will be predicting the cumulative number of confirmed COVID19 cases in various locations across the world, as well as the number of resulting fatalities, for future dates.

We understand this is a serious situation, and in no way want to trivialize the human impact this crisis is causing by predicting fatalities. Our goal is to provide better methods for estimates that can assist medical and governmental institutions to prepare and adjust as pandemics unfold.

File descriptions

- train.csv - the training data up to Mar 18, 2020.

- test.csv - the dates to predict; there is a week of overlap with the training data for the initial Public leaderboard. Once submissions are paused, the Public leaderboard will update based on last 28 days of predicted data.

- submission.csv - a sample submission in the correct format; again, predictions should be cumulative

---

**Generation**

Role: Assistant

Assistant: To tackle this task, we will follow these steps:

Load the Data: Read the training and test datasets.

Preprocess the Data: Handle missing values, convert date columns to datetime objects, and perform any necessary feature engineering.

Model Training: Train a model to predict the cumulative number of confirmed COVID-19 cases and fatalities.

Make Predictions: Use the trained model to make predictions on the test dataset.

Save Predictions: Save the predictions in the required format to a CSV file.

Let's start by implementing the solution in Python:

```python
# filename: covid19_forecasting.py
import pandas as pd
from sklearn.linear_model import LinearRegression
from sklearn.preprocessing import LabelEncoder
import numpy as np
# Load the data
train_path = './covid19-global-forecasting-week-1/train.csv'
test_path = './covid19-global-forecasting-week-1/test.csv'
submission_path = './covid19-global-forecasting-week-1/submission.csv'
output_path = './covid19-global-forecasting-week-1.csv'
train = pd.read_csv(train_path)
test = pd.read_csv(test_path)
submission = pd.read_csv(submission_path)
# Preprocess the data
def preprocess_data(df):
    df['Date'] = pd.to_datetime(df['Date'])
    df['Year'] = df['Date'].dt.year
    df['Month'] = df['Date'].dt.month
    df['Day'] = df['Date'].dt.day
    df = df.drop(columns=['Date'])
    return df
train = preprocess_data(train)
test = preprocess_data(test)
# Encode categorical features
le = LabelEncoder()
train['Province_State'] = le.fit_transform(train['Province_State'].astype(str))
train['Country_Region'] = le.fit_transform(train['Country_Region'].astype(str))
test['Province_State'] = le.transform(test['Province_State'].astype(str))
test['Country_Region'] = le.transform(test['Country_Region'].astype(str))

# Prepare the data for modeling
X_train = train.drop(columns=['ConfirmedCases', 'Fatalities'])
y_train_cases = train['ConfirmedCases']
y_train_fatalities = train['Fatalities']
X_test = test.drop(columns=['ForecastId'])

# Train the model for Confirmed Cases
model_cases = LinearRegression()
model_cases.fit(X_train, y_train_cases)

# Train the model for Fatalities
model_fatalities = LinearRegression()
model_fatalities.fit(X_train, y_train_fatalities)

# Make predictions
predictions_cases = model_cases.predict(X_test)
predictions_fatalities = model_fatalities.predict(X_test)
# Prepare the submission file
submission['ConfirmedCases'] = predictions_cases
submission['Fatalities'] = predictions_fatalities
# Ensure no negative predictions
submission['ConfirmedCases'] = submission['ConfirmedCases'].apply(lambda x:
    ↪ max(0, x))
submission['Fatalities'] = submission['Fatalities'].apply(lambda x: max(0, x))

# Save the submission file
submission.to_csv(output_path, index=False)
print(f"Predictions saved to {output_path}")
```

Please run the above script to generate the predictions and save them to the specified CSV file.
Once the script is executed, the predictions will be saved in the path ./covid19-global-forecasting-week-1.csv.

Table 14: In this sample, we show a data modeling task AotoGen with GPT-4 can generate the submission file.

**Introduction**

Essay writing is an important method to evaluate student learning and performance. It is also time-consuming for educators to grade by hand. Automated Writing Evaluation (AWE) systems can score essays to supplement an educator's other efforts. AWEs also allow students to receive regular and timely feedback on their writing. However, due to their costs, many advancements in the field are not widely available to students and educators. Open-source solutions to assess student writing are needed to reach every community with these important educational tools.

Previous efforts to develop open-source AWEs have been limited by small datasets that were not nationally diverse or focused on common essay formats. The first Automated Essay Scoring competition scored student-written short-answer responses, however, this is a writing task not often used in the classroom. To improve upon earlier efforts, a more expansive dataset that includes high-quality, realistic classroom writing samples was required. Further, to broaden the impact, the dataset should include samples across economic and location populations to mitigate the potential of algorithmic bias.

In this competition, you will work with the largest open-access writing dataset aligned to current standards for student-appropriate assessments. Can you help produce an open-source essay scoring algorithm that improves upon the original Automated Student Assessment Prize (ASAP) competition hosted in 2012?

Competition host Vanderbilt University is a private research university in Nashville, Tennessee. For this competition, Vanderbilt has partnered with The Learning Agency Lab, an Arizona-based independent non-profit focused on developing the science of learning-based tools and programs for the social good.

To ensure the results of this competition are widely available, winning solutions will be released as open source. More robust and accessible AWE options will help more students get the frequent feedback they need and provide educators with additional support, especially in underserved districts.

**Evaluation**

Submissions are scored based on the quadratic weighted kappa, which measures the agreement between two outcomes. This metric typically varies from 0 (random agreement) to 1 (complete agreement). In the event that there is less agreement than expected by chance, the metric may go below 0.

The quadratic weighted kappa is calculated as follows. First, an N x N histogram matrix O is constructed, such that $O_{i,j}$ corresponds to the number of essay_ids i (actual) that received a predicted value j. An N-by-N matrix of weights, w, is calculated based on the difference between actual and predicted values:

$$w_{i,j} = \frac{(i-j)^2}{(N-1)^2}.$$ 

(2)

An N-by-N histogram matrix of expected outcomes, E, is calculated assuming that there is no correlation between values.

This is calculated as the outer product between the actual histogram vector of outcomes and the predicted histogram vector, normalized such that E and O have the same sum.

From these three matrices, the quadratic weighted kappa is calculated as:

$$\kappa = 1 - \frac{\sum_{i,j} w_{i,j} O_{i,j}}{\sum_{i,j} w_{i,j} E_{i,j}}. \tag{3}$$

Submission File For each essay_id in the test set, you must predict the corresponding score (described on the Data page). The file should contain a header and have the following format:

```
essay_id,score
000d118,3
000fe60,3
001ab80,4
...
```

**Data**

The competition dataset comprises about 24000 student-written argumentative essays. Each essay was scored on a scale of 1 to 6 (Link to the Holistic Scoring Rubric). Your goal is to predict the score an essay received from its text.

File and Field Information

- train.csv - Essays and scores to be used as training data.
  - essay_id - The unique ID of the essay
  - full_text - The full essay response
  - score - Holistic score of the essay on a 1-6 scale
- test.csv - The essays to be used as test data. Contains the same fields as train.csv, aside from exclusion of score. (Note: The rerun test set has approximately 8k observations.)
- sample_submission.csv - A submission file in the correct format.
  - essay_id - The unique ID of the essay
  - score - The predicted holistic score of the essay on a 1-6 scale

Please note that this is a Code Competition.

**Generation**

Role: User

To tackle the task of predicting essay scores using the provided dataset, I will outline a plan and then provide you with the Python code to execute. The plan involves the following steps:
1. Data Loading: Load the training and test datasets. 2. Data Preprocessing: Prepare the data for modeling, which might include cleaning the text data.
3. Feature Engineering: Convert the essay texts into a numerical format that can be used by machine learning models, using techniques like TF-IDF.
4. Model Training: Train a machine learning model using the training data. Given the nature of the task, a regression model or a specialized text model like BERT could be appropriate.
5. Prediction: Use the trained model to predict scores for the essays in the test dataset.
6. Output: Save the predictions in the specified CSV format and location.

Here is the Python code that encapsulates these steps. You should run this code in your Python environment where the data files are located.

```python
    # filename: essay_scoring_prediction.py

import pandas as pd
from sklearn.feature_extraction.text import TfidfVectorizer
from sklearn.ensemble import RandomForestRegressor
from sklearn.model_selection import train_test_split
from sklearn.metrics import mean_squared_error
import os

# Load data
train_path = './learning-agency-lab-automated-essay-scoring-2/train.csv'
test_path = './learning-agency-lab-automated-essay-scoring-2/test.csv'
train_data = pd.read_csv(train_path)
test_data = pd.read_csv(test_path)

# Preprocess data
tfidf = TfidfVectorizer(stop_words='english', max_features=5000)
train_features = tfidf.fit_transform(train_data['full_text'])
test_features = tfidf.transform(test_data['full_text'])

# Model training
model = RandomForestRegressor(n_estimators=100)
model.fit(train_features, train_data['score'])

# Prediction
predictions = model.predict(test_features)
predictions = [round(score) for score in predictions] # Round scores to the
    ↪ nearest integer

# Prepare submission file
submission = pd.DataFrame({
   'essay_id': test_data['essay_id'],
   'score': predictions
})

# Save submission file
output_path = './learning-agency-lab-automated-essay-scoring-2.csv'
submission.to_csv(output_path, index=False)

print(f"Submission file saved to {output_path}")
```

This script uses a RandomForestRegressor for prediction, which is a simple yet effective model for regression tasks. The TF-IDF vectorizer is used to convert text data into a format suitable for machine learning. After running this script, the predictions will be saved in the specified output directory. Make sure to check the output directory for the results.

Role: Assistant

Exit code: 124 (execution failed)
Code output:

```
    Timeout
```

User: Assistant

The timeout error suggests that the process took too long to execute, likely due to the size of the data and the complexity of the operations (such as TF-IDF transformation and RandomForest training). To address this, we can simplify the model and reduce the computational load. Here's an updated approach:
1. Reduce Feature Space: Decrease the number of features in the TF-IDF vectorizer.

2. Simplify Model: Use a simpler model that requires less computation time, such as a linear regression model.

Here's the revised Python code with these adjustments:

```python
from sklearn.feature_extraction.text import TfidfVectorizer
from sklearn.linear_model import LinearRegression
import os

# Load data
train_path = './learning-agency-lab-automated-essay-scoring-2/train.csv'
test_path = './learning-agency-lab-automated-essay-scoring-2/test.csv'
train_data = pd.read_csv(train_path)
test_data = pd.read_csv(test_path)

# Preprocess data
tfidf = TfidfVectorizer(stop_words='english', max_features=1000) # Reduced
    ↪ number of features
train_features = tfidf.fit_transform(train_data['full_text'])
test_features = tfidf.transform(test_data['full_text'])

# Model training
model = LinearRegression()
model.fit(train_features, train_data['score'])

# Prediction
predictions = model.predict(test_features)
predictions = [round(score) for score in predictions] # Round scores to the
    ↪ nearest integer

# Prepare submission file
submission = pd.DataFrame({
    'essay_id': test_data['essay_id'],
    'score': predictions
})

# Save submission file
output_path = './learning-agency-lab-automated-essay-scoring-2.csv'
submission.to_csv(output_path, index=False)

print(f"Submission file saved to {output_path}")
```

This version of the script uses a Linear Regression model, which is computationally less intensive than RandomForest, and reduces the number of features in the TF-IDF vectorizer to speed up the text processing. Please run this updated script and check if it completes within the allowed time.

Role: Assistant
Exit code: 0 (execution succeeded)
Code output:

```
Submission file saved to ./learning-agency-lab-automated-essay-scoring-2.csv
```

Role: User
Great! The script executed successfully and the submission file has been saved to the specified location. You can now check the output file to ensure that it contains the predicted scores formatted correctly according to the competition's requirements.

If everything looks good, you can proceed with any further steps required by the competition, such as submitting the results or performing additional analysis.

If you need further assistance or have more tasks, feel free to ask. Otherwise, good luck with your competition entry!

