# OpenReview forum: "DSBench: How Far Are Data Science Agents from Becoming Data Science Experts?"
_ICLR.cc/2025/Conference — ICLR 2025 Poster_

### Official Review · Reviewer_XL8J · 2024-10-27

**Soundness:** 3
**Presentation:** 2
**Contribution:** 3
**Rating:** 6
**Confidence:** 4

**Summary:**

The paper constructs DSBench, a benchmark focused on evaluating LM-based data science agents through realistic tasks sourced from ModelOff and Kaggle competitions.

**Strengths:**

1. I believe the dataset provided by the authors can, to some extent, reflect a model's ability to tackle data science tasks, and even today, it remains quite challenging, serving as a non-trivial evaluation benchmark.
2. The authors conducted some interesting analyses across various dimensions for different models and different data science tasks (e.g., examining task completion rates relative to task release time, the correlation between task difficulty and context length, etc.), offering potential insights into the development of LM as agents.

**Weaknesses:**

1. **Data Source:**
- All the data analysis tasks mentioned by the authors at line 106 are related to finance. I would appreciate it if the authors could clarify this point in the paper and explain why other types of analysis tasks are not as suitable. This concern arises because ModelOff is actually a global financial modeling competition.
- Similarly, for data modeling, it seems this is also influenced by the fact that there are numerous modeling task competitions on Kaggle.

I wonder if the authors have explored more platforms or data sources and could explain why they were not suitable for evaluating data science agents?
Additionally, in my opinion, competitions are not always the closest representation of the real world. For example, as far as I know, Spider2-V[1] incorporates a lot of tools and software from industrial data pipelines. Could this be a more realistic measure of real-world scenarios?

2. **Evaluation Metrics:**
- I fully understand that collecting and building complex evaluation environments is a considerable engineering task. However, if the evaluation is based solely on competition platforms and existing output-only metrics, it seems that it may not fully capture the comprehensive capabilities of data science agents. This is similar to what the authors mentioned in lines 153 and 154, such as extracting insights, proper data handling, etc.

3. **Need better presentation, especially for some tables and figures:**
- I noticed in Figure 4, some of the models do not have the reported accuracy (does this mean zero accuracy?); while the width of the bars are not set as the same. However, I do not find clear explanation to these.
- I believe transforming Table 5 into a line chart with corresponding accuracy values would clearly illustrate the trend in accuracy over time.

[1] Spider2-V: How Far Are Multimodal Agents From Automating Data Science and Engineering Workflows?, NeurIPS 2024.

**Questions:**

1. As highlighted in Table 1, What is the purpose of distinguishing tables from data files? What is the difference between tables and data files? I would like the authors to clarify how they treat these two types of data samples during evaluation.

2. I also find the taxonomy somewhat difficult to understand. For example, "tables" and "excels" are categorized separately, and I hope the author can clarify the distinctions between these categories more clearly.

---

> ### Author Response · Authors · 2024-11-20
>
> > **Data Source and Evaluation.**
> - Thank you for your comments. To clarify, the majority of tasks in ModelOff are finance-related and financial analysis constitutes a substantial portion of real-world data analysis tasks. Financial modeling tasks often serve as representative benchmarks due to their complexity, requiring sophisticated reasoning, integration of multimodal data (e.g., spreadsheets, textual descriptions), and precise quantitative analysis. These characteristics make financial analysis a particularly suitable domain for evaluating the capabilities of data science agents. Additionally, while ModelOff primarily focuses on finance, **it also includes tasks from other areas, such as election statistics, adding some diversity to the benchmark**. We will revise the manuscript to explicitly highlight this rationale.
> - For data modeling, Kaggle was selected due to its status as the most popular platform for data science competitions, covering a broad range of domains such as healthcare, retail, and environmental science. While other platforms and data sources can indeed be utilized, Kaggle's extensive task diversity and established reputation make it a robust choice for constructing our benchmark.
> - Regarding the use of industrial pipelines, such as those incorporated in Spider2-V, we agree that these represent realistic workflows. However, our benchmark deliberately avoids restricting the tools or systems that can be used. This design decision provides greater freedom for participants and encourages creative approaches to problem-solving. In contrast, benchmarks like Spider2-V emphasize LLM's tool-specific capabilities in controlled environments, which are excellent for evaluating system-specific proficiencies but differ from our goal of focusing on evaluating performance of different systems.
> - We will clarify these points in the revised manuscript and appreciate your feedback in helping us improve the paper’s scope and explanation.
>
>
> > **Better Presentation.**
> - Thank you for your valuable suggestions. Regarding Figure 4, the absence of reported accuracy for some models indeed indicates zero accuracy. The varying widths of the bars have no special significance, and I will add explanations to clarify this in the paper.
> - As for Table 5, we agree that transforming it into a line chart with corresponding accuracy values would better illustrate the trend in accuracy over time. I will make this adjustment to enhance the clarity and visual presentation of the data.
>
>
> > **Difference between tables and data files in Table 1.**
> - We apologize for any misunderstanding caused. In our context, "table" typically refers to data presented directly in the task instruction in textual form. These tables are generally simpler and contain smaller amounts of data. In contrast, "data files" refer to larger datasets provided in formats such as CSV or Excel. For evaluation, small tables in textual form can be directly concatenated into the prompt for processing. On the other hand, data files may either be concatenated into the prompt (if feasible) or require the agent to write code to interact with and process the file. We will clarify these points in our paper.
>
> > **I also find the taxonomy somewhat difficult to understand. For example, "tables" and "excels" are categorized separately, and I hope the author can clarify the distinctions between these categories more clearly.**
> - Thank you for pointing this out. We understand that the distinction in our taxonomy may not be immediately clear and will revise the manuscript to provide better clarification. "Tables" refer to structured data presented directly within the task instruction in textual format. These are typically smaller datasets and can be processed directly within the prompt. "Excels" refer to larger, external data files provided in formats like xlsx file, which may require agents to write code for processing due to their size and complexity. We will update the manuscript to ensure this distinction is explicitly stated.

---

> > ### Author Response · Authors · 2024-11-21
> >
> > Thank you for your valuable feedback. As the ICLR public discussion phase will be ending in a few days, I wanted to check if my previous response has addressed your concerns. If it has, I would greatly appreciate it if you could consider updating your score accordingly. Thank you again for your time and insights!

---

> ### Comment · Reviewer_XL8J · 2024-11-22
> **Official Comment by Reviewer XL8J**
>
> Thanks for the response:
>
> 1. Regarding the differences between tables and Excel, as well as clarifications on other visualizations, I believe the authors have addressed some of my concerns.
>
> 2. However, I still have reservations about the choice of source data and the evaluation metrics. I hope the authors can specifically address the issues related to **Weakness 2**.
>
> **Specifically:**
> If you believe the overall score fully reflects what you mentioned about "extracting insights" and "proper data handling," could you provide relevant examples and analysis to support this?  If not, I strongly recommend that the authors include a discussion of the limitations for current evaluation metrics.
> Overall, I have seen the authors provide a series of explanations, but they have not yet made adjustments in the paper or included necessary discussions and explanations about the aforementioned limitations.
>
> I will still hold my original score as I think this paper is right on the edge of being acceptable, and I do not think I will raise my score further since it is high enough.
> But I really hope the authors could make the necessary revisions, as I believe these adjustments are not particularly difficult to implement.

---

> > ### Author Response · Authors · 2024-11-22
> >
> > Thank you for your thoughtful feedback. We appreciate your acknowledgment that some of your concerns regarding visualizations and the differences between tables and Excel have been addressed. We agree with your point, which includes a discussion of the limitations of current evaluation metrics. We will add a limitation section in our paper to discuss these.
> >
> > Thank you again for your feedback and encouragement in making these necessary revisions. Although you have decided to keep your original score, we greatly value your suggestions and will incorporate them into the revised version. Your comments have been invaluable in improving the clarity and robustness of our work.

---

### Official Review · Reviewer_TVnT · 2024-10-31

**Soundness:** 4
**Presentation:** 3
**Contribution:** 4
**Rating:** 8
**Confidence:** 4

**Summary:**

This paper introduces DSBench, a comprehensive data science benchmark containing 466 data analysis tasks and 74 data modeling tasks sourced from ModelOff and Kaggle competitions. Compared to existing benchmarks, DSBench provides a more realistic evaluation environment, encompassing long-context understanding, multimodal task backgrounds, large data file processing, and multi-table structure reasoning. Through evaluation of state-of-the-art LLMs, LVLMs, and agents, the study finds that they still struggle with most tasks, with the best agent achieving only 34.12% accuracy in data analysis tasks and a 34.74% Relative Performance Gap (RPG) in data modeling tasks.

**Strengths:**

### 1. Originality & Vision
- Pioneering creation of a comprehensive real-world data science benchmark, analogous to SWE-bench in software engineering, marking a significant step toward AGI evaluation
- First benchmark to integrate both data analysis and modeling tasks in realistic settings with complex data structures and multimodal contexts
- Novel introduction of RPG (Relative Performance Gap) metric that effectively normalizes performance across diverse modeling tasks
- Innovative approach to testing both pure language understanding and tool utilization capabilities
- Make consideration of not only the performance but also the cost

### 2. Technical Robustness
- Rigorous task collection methodology, carefully curated from established platforms (ModelOff and Kaggle) ensuring real-world applicability
- Comprehensive coverage of data science tasks: 466 analysis tasks + 74 modeling tasks, spanning different complexity levels and domains
- Sophisticated evaluation framework that considers:
  * Multi-table reasoning capabilities
  * Long-context understanding
  * End-to-end solution generation
  * Tool integration and utilization
  * Multiple modalities processing

### 3. Practical Significance
- Direct application to real-world data science scenarios, bridging the gap between academic benchmarks and practical challenges
- Clear identification of current AI systems' limitations in data science tasks:
  * Understanding complex data relationships
  * Handling multi-step reasoning
  * Managing tool interactions
- Provides valuable insights for developing more capable data science agents by revealing specific areas where current models fall short
- Sets a new standard for evaluating AI systems' practical data science capabilities, essential for progressing toward AGI

### 4. Research Impact
- Creates a foundation for systematic evaluation of data science capabilities in AI systems
- Enables quantitative comparison of different AI approaches in real-world data science scenarios
- Provides a roadmap for developing more capable AI systems that can handle complex, real-world data science tasks
- Serves as a crucial benchmark for measuring progress toward AGI in the domain of data science

**Weaknesses:**

### 1. Statistical Rigor in Dataset Scale
- While 540 samples is reasonable given the scarcity of high-quality ModelOff & Kaggle competitions, the paper could benefit from:
  * Reporting confidence intervals through multiple experimental runs
  * Conducting bootstrap analysis to estimate the robustness of performance metrics
  * Providing power analysis to justify the sample size
- The paper could discuss how the current sample size was determined and what would be an ideal size for future extensions

### 2. Evaluation Methodology
- Human baseline performance (22 competitions) could be strengthened by:
  * Including more expert evaluators per task
  * Reporting inter-rater agreement scores
  * Documenting the selection criteria for human experts
- The evaluation process could benefit from:
  * Explicit discussion of potential biases in task selection
  * Analysis of task difficulty distribution
  * More detailed failure case studies with expert annotations

**Questions:**

In Table 1's comparison section, I suggest reversing the criteria for "Exec. Eval.", "Code only", and "Env. Fix." to negative statements. This way, checkmarks would indicate DSBench's unique advantages where other benchmarks fall short, making it easier for readers to quickly grasp DSBench's contributions. This would better highlight how DSBench addresses limitations in existing benchmarks and make the comparison more intuitive.

---

> ### Author Response · Authors · 2024-11-20
>
> > **Statistical Rigor in Dataset Scale**
> - Thank you for this insightful comment. We appreciate the suggestion to enhance the statistical rigor of our analysis. Regarding the sample size, 540 tasks was determined based on the availability of high-quality competitions from ModelOff and Kaggle and the evaluation cost of evaluated models, offering realistic and diverse data science challenges.
> - To ensure the reliability of our current results, we specifically designed the inference phase to minimize randomness. For all experiments, we use greedy decoding with the temperature set to 0 and top_p set to 1, which reduces variability and ensures consistent results across tasks.
> - We acknowledge that incorporating confidence intervals, bootstrap analysis, and power analysis could further enhance the robustness of our findings. In future work, we plan to conduct multiple experimental runs and report confidence intervals for performance metrics.
>
>
> > **Evaluation Methodology**
> - Thank you for your insightful question. We agree that the human baseline performance could be further strengthened by incorporating more expert evaluators for each task. However, since expert annotation is both time-consuming and expensive, we sampled ten tasks for human evaluation, with each task being annotated by a single expert. The experts were selected based on their qualifications, specifically being either master's graduates in the data science track within computer science or current PhD students specializing in data science. In the future, we will recruit more volunteers to work on annotation.
> - Regarding the evaluation process, we recognize the value of discussing potential biases in task selection and analyzing the distribution of task difficulty. We present the difficulty distribution in Sections 3.2.1 and 3.2.2. In the revised manuscript, we will add a more detailed discussion on these aspects, such the task sucess rate of the human for each task.
>
>
> > **Suggest reversing the criteria for "Exec. Eval.", "Code only", and "Env. Fix." to negative statements.**
> - Thank you for your valuable comments. We will include this comment into our paper.

---

### Official Review · Reviewer_EZxL · 2024-11-02

**Soundness:** 3
**Presentation:** 3
**Contribution:** 3
**Rating:** 8
**Confidence:** 4

**Summary:**

DSBench introduces a benchmark to test the performances of LLM or VLM agents. It contains data analysis and modeling tasks sourced from Kaggle and ModelOff competitions. Compared to other similar benchmarks, the benchmark is more realistic in that it contains data files, tables, and images and provides executable evaluation functions to validate the answers. The paper tests various models and agent systems and found them to pale in comparison to human data scientists.

**Strengths:**

The paper collects a set of data science challenges from Kaggle and ModelOff. By releasing the code and data, the community can use it to measure the performances of their agent tools. The paper demonstrates through experiments that there exists huge gaps between data science agents and humans, demonstrating that the benchmark is not saturated.

**Weaknesses:**

In general, the paper is well written. The only concerns relate to the sustainability of the benchmark; by releasing the dataset, there is a high chance that it will be, either intentionally or unintentionally, incorporated into the training data for LLMs or VLMs. It seems like this possibility has not been considered by the authors and their explanation for the correlation between accuracy and year of release (Table 5, L416-L421) is weak. Some critical information is missing and it affects my judgement. I would be happy to change my score if the authors can clear my potential misunderstandings.

**Questions:**

1. In Table 3, what does "context length" refer to? Is it the number of English words, characters, or tokens?
2. The paper assumes that the LLMs should be able to generate the answers from the data files. Is it possible for the LLMs to know the answers to the questions from pretraining? For example, the answers to the challenge may be discussed in a Reddit forum that has been scraped in pretraining.
3. In Figure 2(b), what are A-F, A-I, and A-D?
4. L265: what is N in \hat{F} = \mathcal{G}(E,N,S,M)?
5. Figure 3: Can you define m0, m1, ..., m17, perhaps as a separate table in the Appendix?
6. Please define the versions of the model used in the main paper instead of relegating it in the Appendix.
7. What are the settings for the models (e.g., temperature)? Are the experiments conducted multiple times? If so, how are they aggregated?
8. For the analyses (e.g., L404 -- L421), it would be nice to state if they have been corroborated by prior research. For example, Qian et al. has also observed that ML agents perform better on older Kaggle challenges and hypothesize that older challenges have more data leakage into pretraining data.
9. Can you provide a unified script to download the datasets, perhaps using an API, process the downloaded data and run the experiments? It would help for readers to verify the results.
10. L406-L409: Figure 4 shows only GPT-4o, AutoGen and Gemini, but your text references Llama3-8B and GPT 3.5.
11. How do I interpret the RPG of human? Doesn't 'human' represent the best human generated answer and RPG compares against the best human?

---

> ### Author Response · Authors · 2024-11-20
>
> > **Concern of sustainability of the benchmark and more explanation for Table 5.**
> - Thank you for your comment. We agree that there is a chance that our benchmark could be incorporated into the training data for LLMs or VLMs, either intentionally or unintentionally. To address this, we plan to implement strict licensing (as shown in the following) prohibiting its use in training. Additionally, these data analysis tasks were released before the era of large-scale LLMs (models with over 7 billion parameters), so the low accuracy may indicate that current models do not yet contain these tasks in their training data. (Because if they had been trained on these data, their performance might be significantly higher.)
> - DSBench Usage Terms
> The use of DSBench is strictly limited to research and evaluation purposes. Any use of the benchmark or its data for training machine learning models, including large language models (LLMs) and vision-language models (VLMs), is strictly prohibited. Redistribution for non-commercial research is permitted with proper attribution. Unauthorized use, including for model training, may result in legal action.
> For inquiries or permissions, please contact: xxxxx.
>
> > **More explanation for Table 5.**
> - In Table 5, we observe that the difficulty of the challenges increases over the years. This trend can be attributed to the evolution of data technology and the escalating complexity of the questions. To verify this, we further show the human performance over different years, which demonstrates a similar trend with GPT-4o, i.e., the accuracy decreases as the years progress.
> - We will add these explanations in our paper. If you still have confusion, please tell us.
>
> > **Q1: In Table 3, what does "context length" refer to? Is it the number of English words, characters, or tokens?**
> - Thanks for your question. It's number of English words of the problem.
>
> > **Q2: Is it possible for the LLMs to know the answers to the questions from pretraining? For example, the answers to the challenge may be discussed in a Reddit forum that has been scraped in pretraining.**
> - Thanks for your valuable comment. Because of the low performane of the existing models, we believe that these tasks are unlikely to be included in existing pre-training data. In addition, the data collection strategy is general and can be used to collect new data modeling tasks.
>
> > **Q3:In Figure 2(b), what are A-F, A-I, and A-D?**
> - Thank you for your question regarding Figure 2(b). To clarify, in the figure, "A-F" indicates that the question contains five answer options, labeled as A, B, C, D, and F. Similarly, "A-I" represents questions with nine options, ranging from A to I, and "A-D" corresponds to questions with four options, labeled as A, B, C, and D. We will add this clarification to the manuscript.
>
> > **Q4: L265: what is N in \hat{F} = \mathcal{G}(E,N,S,M)?**
> - Thank you for catching this typo. We confirm that in
> \hat{F} = \mathcal{G}(E,N,S,M), "N" should indeed be "A". We will correct this in the revised manuscript.
>
> > **Q5: Figure 3: Can you define m0, m1, ..., m17, perhaps as a separate table in the Appendix?**
> - Thanks for your advice. Of course, we will add a Table in the Appedx for the defination of m0, m1, ..., m17 as follows.
> -
> | Metric Code | Metric Name                                   |
> |-------------|----------------------------------------------|
> | m0          | Accuracy                                     |
> | m1          | ROC                                          |
> | m2          | Normalized Gini Coefficient                 |
> | m3          | Root Mean Squared Logarithmic Error         |
> | m4          | Root Mean Squared Error                     |
> | m5          | R2 Score                                    |
> | m6          | Mean Columnwise Root Mean Squared Error     |
> | m7          | Macro F1                                    |
> | m8          | Micro-averaged F1                           |
> | m9          | Mean Absolute Error                         |
> | m10         | Word-level Jaccard Score                    |
> | m11         | Quadratic Weighted Kappa                    |
> | m12         | Pearson Correlation Coefficient             |
> | m13         | Median Absolute Error                       |
> | m14         | Symmetric Mean Absolute Percentage Error    |
> | m15         | Mean Column-wise Spearman’s Correlation Coefficient |
> | m16         | MPA@3                                       |
> | m17         | Logarithmic Loss                            |

---

> > ### Author Response · Authors · 2024-11-20
> >
> > > **Q6: Please define the versions of the model used in the main paper instead of relegating it in the Appendix.**
> > - Thanks for your suggestion. We will remove the version of model into Section 3.3 of the main paper.
> >
> > > **Q7: What are the settings for the models (e.g., temperature)? Are the experiments conducted multiple times? If so, how are they aggregated?**
> > - Thank you for your question. In all experiments, we use greedy decoding for all models, with the temperature set to 0 and top_p set to 1. This configuration minimizes randomness in the output and ensures reproducibility. Additionally, we perform inference only once for all models. This decision not only aligns with our parameter settings to mitigate randomness but also helps reduce computational costs. We will clarify this in the revised manuscript.
> >
> > > **Q8: For the analyses (e.g., L404 -- L421), it would be nice to state if they have been corroborated by prior research. For example, Qian et al. has also observed that ML agents perform better on older Kaggle challenges and hypothesize that older challenges have more data leakage into pretraining data.**
> > - Thanks for your advice. We will add more analysis in  L404 -- L421. 1) We found that the performance of agents declines as the total context length increases. This observation aligns with findings in other tasks for agents, such as SWEBench, which reports that model performance drops considerably for bug-fixing tasks as the total context length increases.  2) We also observe that the difficulty of challenges increases over the years. This observation is consistent with existing research. For example, Qian et al. has also observed that ML agents perform better on older data science challenges.
> >
> > > **Q9: Can you provide a unified script to download the datasets, perhaps using an API, process the downloaded data and run the experiments? It would help for readers to verify the results.**
> > - Thanks for your advice. We will add a unified script for data dowloading and experiment running.
> >
> > > **Q10: L406-L409: Figure 4 shows only GPT-4o, AutoGen and Gemini, but your text references Llama3-8B and GPT 3.5.?**
> > - Thanks for your question. We only show the performance of GPT-4o, AutoGen, and Gemini in Figure 4 due to space constraints in the figure. However, we observed similar trends across other models, such as Llama3-8B and GPT-3.5. We will include this observation in the appendix to provide a more comprehensive discussion.
> >
> > > **Q11: How do I interpret the RPG of human? Doesn't 'human' represent the best human generated answer and RPG compares against the best human?**
> > - Thank you for your comment. You are correct that the RPG metric compares performance against the best possible result. Since it is difficult to obtain solutions from the top-performing individuals for every competition, we instead use the maximum achievable score for each competition's evaluation metric (i.e., the perfect score) as the benchmark. In this context, the best result corresponds to an RPG of 100%. This approach ensures consistency and provides a clear standard for comparison across all tasks. We will clarify this in the revised manuscript.

---

> > > ### Author Response · Authors · 2024-11-21
> > >
> > > Thank you for your valuable feedback. As the ICLR public discussion phase will be ending in a few days, I wanted to check if my previous response has addressed your concerns. If it has, I would greatly appreciate it if you could consider updating your score accordingly. Thank you again for your time and insights!

---

> > > ### Comment · Reviewer_EZxL · 2024-11-22
> > >
> > > Thank you for your responses to my questions. I believe that most of the major concerns have been addressed. However, I remain confused about the definition of "RPG of humans" in Table 6. Can you explain the steps of how it is calculated?
> > >
> > > I also feel that there is insufficient awareness of the possibility of data leakage into pretraining. Data leakage can occur in many ways beyond just having the full description and solution being part of the pretraining; it can also occur when partial questions and answers are discussed on online forums or when the original problems are reframed and the solutions are posted (as is typical on stackoverflow). Older data challenges should have more discussion threads and have established methods (at the current point in time) whereas newer ones have less discussion threads and may require more novel approaches. At the very least, this should be pointed out as a possible explanation for some of your observations. I would also encourage the authors to think more deeply about how it may limit the utility of the benchmark and discuss how to mitigate this issue beyond putting a disclaimer or an easily ignored license.

---

> ### Author Response · Authors · 2024-11-22
> **Thanks for your further valuable feedback**
>
> Thanks for your further valuable feedback. We answer your question as follows.
> > **Steps of how RPG is calculated**
> - Thank you for your comment. We compute RPG of human as follows, $1/N \sum_{i=1}^N \times max((h_i-b_i)/(g_i-b_i))$, where $N$ is the total number of competitions. $b_i$ is the performance of the baseline. $h_i$ is the human performance. $g_i$ is the highest performance value for the $i$-th competition. For example,  $g_i$ is 1 when the metric of the competition is accuracy, $g_i$ is 0 when the metric of the competition is MSE (Mean Squared Error).
>
> > **Possible data leakage.**
> - This is a good question since all existing benchmarks suffer the same quesiton. As noted in Table 5 of our paper, we observe a clear trend where tasks from older challenges yield different performance compared to newer ones, potentially supporting the hypothesis of varying exposure. We will explicitly discuss this in the revision to make the possible effects of leakage clearer.
> - In addition, we are exploring the inclusion of entirely new data challenges in future iterations of DSBench to evaluate models without the risk of prior exposure. While we understand that a license alone may not completely prevent misuse, it sets a clear legal framework against incorporating DSBench into training datasets.
>
> Thanks again for your good questions.

---

> > ### Comment · Reviewer_EZxL · 2024-11-22
> >
> > Thank you for your responses. There seems to be a minor mistake in the equation, but I think I figured it out. Who are the human expert(s) mentioned in the paper and how are they recruited? Why do they score much worse than the best performance value for the competitions (who presumely are also humans)? Please consider adding information about the human expert.
> >
> > I do not have further questions. Thanks for taking the lead on benchmarking DS agents.

---

> > > ### Author Response · Authors · 2024-11-22
> > >
> > > Thank you for your thoughtful feedback. For the human expert performance in our study, we collected and evaluated the top-3 performing codes from the Kaggle website for each competition (e.g., https://www.kaggle.com/competitions/santa-2024/code). The human expert performance on our split dataset is then calculated as the average performance of these top-3 codes.
> > > Regarding the "best performance" metric, this is not derived from a human evaluator. Instead, we use the maximum achievable score based on the competition's evaluation metric as the "best performance." This represents the theoretical upper bound for the competition. As a result, the human expert performance is naturally lower than this best performance value since it is practically impossible to consistently achieve the maximum score across all competitions.
> > >
> > > We appreciate your suggestion to include more information about the human expert evaluations in the paper, and we will incorporate these clarifications in the revised version. Thank you again for your detailed review and for supporting efforts in benchmarking DS agents.
> > >
> > > If you have no further questions, **we kindly ask if you could consider adjusting the score** in light of our responses. Thank you again for your detailed review and for supporting efforts in benchmarking DS agents.

---

> > > > ### Comment · Reviewer_EZxL · 2024-11-22
> > > >
> > > > After much consideration, I will maintain my score.
> > > >
> > > > This work is relevant and will be helpful to the community. The experiments make sense from a high level and the findings are consistent with prior work. The authors have been responsive over the course of the public discussion phase. I would have given this paper a 7 if I have the option to.
> > > >
> > > > However, I can't recommend an 8 (clear accept) as I am still not convinced about the benchmark's resistance to data leakage, which affects the reliability of the results and the durability of the benchmark.

---

> > > > > ### Author Response · Authors · 2024-11-22
> > > > >
> > > > > Thank you for your thoughtful and constructive feedback. We greatly appreciate your recognition of the relevance and impact of our work, as well as your acknowledgment of our responsiveness during the discussion phase.
> > > > >
> > > > > Regarding your concern about the benchmark's resistance to data leakage, we utilized specific strategies to mitigate this issue for both data modeling and data analysis tasks. For kaggle competitions, we re-split the dataset, keep test set labels only for evaluation, and replace them with dummy values, which is similar to the existing strategy [1]. This prevents agents from directly using memorized solutions or labels. In addition, for all tasks, we prompt the model to output reasoning and interpretation beyond simply recalling memorized responses. This design discourages agents from relying on shortcuts or rote memorization.
> > > > >
> > > > > We hope our response addresses all your concerns.
> > > > >
> > > > > [1] ScienceAgentBench: Toward Rigorous Assessment of Language Agents for Data-Driven Scientific Discovery

---

### Official Review · Reviewer_2rKh · 2024-11-04

**Soundness:** 3
**Presentation:** 3
**Contribution:** 2
**Rating:** 6
**Confidence:** 2

**Summary:**

The paper introduces a new benchmark for automated data science, DSBench.  This
encompasses both excel exercises from ModelOff and kaggle competitions.  DSBench
also includes metrics for evaluating success in both kinds of tasks.  The paper
evaluates several open and closed source LLMs/VLMs.  The key take away is that
SOTA architectures are still far away from achieving human-level performance.

**Strengths:**

**Originality**: This is chiefly an engineering paper, it introduces no novel techniques or ideas.  DSBench collates existing resources.

**Quality**: The key contribution is DSBench, which I am confident took quite some effort to set up.  All in all, this contribution seems to be of good quality: it encompasses a large number of relevant tasks which go beyond what the literature currently offers.  It also comes with some rather natural metrics.  The interesting bit is the evaluation, which on the one hand shows that DSBench can indeed be used as intended, and on the other points out a performance gap for existing architectures.  The findings are otherwise quite intuitive: newer models perform better, more complex tasks are harder to solve.  The real contribution is the research that DSBench will enable.

Unfortunately, as a non-expert (my work on automating data science predates LLMs), I cannot assess the overlap between DSBench and recent works in this area.

**Clarity**: The text is generally readable, with a few linguistic quirks here and there.  The figures are easy to understand.

**Significance**:  Automated data science is a central topic nowadays.  It is not impossible that DSBench will provide a significant boost to auto-DS.  But again, this depends on overlap with existing work, which I am not overly familiar with.  As a result, I have decided to grade conservatively the contribution aspect of the paper.

**Weaknesses:**

**Quality**: The only relatively minor issue I'd like to point out concerns Section 3.3, which promises discussing the errors made by LLMs, but does not provide any details, really.  I think this should be amended.

**Questions:**

Plase see weaknesses.

---

> ### Author Response · Authors · 2024-11-20
>
> > **Novelty techniques or ideas.**
> - We sincerely thank you for your time, effort, and thoughtful comments in reviewing our work. We greatly appreciate the detailed feedback, which has helped us better articulate the contributions of our paper. While DSBench does build upon existing resources like ModelOff and Kaggle, we want to emphasize that it is more than a collection of existing tasks. Our work introduces a realistic and comprehensive benchmark framework, incorporating multimodal tasks, long-context reasoning, and end-to-end evaluations to better reflect real-world data science challenges. Additionally, we propose the Relative Performance Gap metric, a novel method for normalizing diverse evaluation metrics, enabling fair and consistent comparisons. We believe these elements represent meaningful innovations that go beyond engineering contributions and provide valuable insights for advancing data science agents. Thank you again for your thoughtful review and constructive feedback.
>
> > **Different with the existing work.**
> - Thank you for your thoughtful feedback and for sharing your perspective on our work. We understand the difficulty in assessing overlap in a rapidly evolving field like automated data science. To clarify, DSBench is distinct from existing benchmarks due to its focus on **realistic, multimodal, long-context, and end-to-end tasks** that closely mirror real-world challenges, setting it apart from prior work that often emphasizes **simplified tasks like code generation**. Additionally, our introduction of the Relative Performance Gap metric addresses a critical gap in **evaluating diverse tasks by enabling fair and consistent comparisons**. By leveraging tasks from ModelOff and Kaggle, DSBench provides a practical and challenging benchmark that highlights the current limitations of state-of-the-art agents. We hope this clarification underscores DSBench’s significance and unique contributions to the field, and we greatly appreciate your time and consideration in reviewing our work.
>
> > **The only relatively minor issue I'd like to point out concerns Section 3.3, which promises discussing the errors made by LLMs, but does not provide any details, really. I think this should be amended.**
> - Thank you for the suggestion. We show in-depth analysis with case study in Appendix H. In particular, we show 7 testing examples for in-depth analysis. For example, in Figure 9, we show the introduction of the challenge, the question, a screenshot of a portion of the data file, and the generation text from GPT-4o. Based on the generation from GPT-4o, we found that the model misunderstood the meaning of the introduction and misinterpreted the district code as the voter identity code, leading to the wrong answer. We conducted comprehensive error analysis, we will add a reference in Section 3.3 to the appendix.
> - If you have other suggestions pls let us know. If it's the only concern, could you consider increasing the contributions/ratings.

---

> > ### Comment · Reviewer_2rKh · 2024-11-20
> >
> > Thank you for your reply.
> >
> > > **it is more than a collection of existing tasks**
> >
> > sorry, I did not mean to be dismissive of your work.  what I meant is that dsbench collects existing resources and packages them neatly by introducing useful additions.  this was not meant to be a negative comment.
> >
> > > **difference wrt existing work**
> >
> > as I mentioned, I am not expert enough to really judge the novelty wrt existing works.  I am interested in knowing what other reviewers have to say on the subject.
> >
> > > **appendix H**
> >
> > thank you for pointing this out.
> >
> > Rest assured that I will take these clarifications into consideration when submitting the updated score.

---

> > > ### Author Response · Authors · 2024-11-22
> > >
> > > Thank you for your thoughtful feedback and clarifications. We truly appreciate your understanding and constructive comments.
> > >
> > > As for the overall reception, Reviewer TVnT believes our paper is strong and should be accepted, and Reviewer EZxL acknowledges that we have addressed their concerns. In light of this, we kindly ask if you could consider raising your score to reflect the clarifications and additional efforts we’ve provided. Thank you again for your time and consideration!

---

### Official Review · Reviewer_cMgg · 2024-11-08

**Soundness:** 3
**Presentation:** 3
**Contribution:** 2
**Rating:** 6
**Confidence:** 3

**Summary:**

This paper introduces DSBench, a comprehensive data science benchmark designed to assess the performance of data science agents on real-world tasks. DSBench integrates the challenges from ModelOff and Kaggle competitions, creating a realistic environment that covers 466 data analysis and 74 data modeling tasks. To enable fair comparison, the paper proposes the Relative Performance Gap metric, which normalizes various evaluation metrics for data modeling tasks. Evaluation of state-of-the-art models, including GPT-4o, Claude, and Gemini, reveals that DSBench remains challenging, with significant performance gaps between these models and human capabilities.

**Strengths:**

**Novelty**
- This paper offers a fresh contribution to data science by introducing DSBench, a new benchmark that evaluates data science agents under realistic task conditions derived from ModelOff and Kaggle competitions. It pushes the boundaries of traditional benchmarking.

**Quality**
- The design of DSBench, with its comprehensive task types and Relative Performance Gap (RPG) metric, demonstrates rigor in addressing evaluation inconsistencies across various modeling tasks.

**Clarity**
- Task designs, methods, and performance comparisons are clear and well-organized. They contain many details but are not hard to follow.

**Significance**
- DSBench sets a new standard in evaluating data science agents, driving advancements in realistic, end-to-end task performance. Its contributions are useful to future advancements of intelligent, autonomous data science agents.

**Weaknesses:**

- I'm bit unsure of the robustness and persuasiveness of the RPG metric is valid. It could be beneficial to further assess how well the RPG reflects actual performance across varying data types and task complexities.
- Although Kaggle tasks are highly relevant, they often focus on a narrow subset of domains (e.g., retail, finance).

**Questions:**

1. Can authors please explain if the metric and dataset are sufficient to cover the real-world diversity that the benchmark aims to address?
2. Explain why RPG is able to reflect actual performance across varying data types and task complexities.

---

> ### Author Response · Authors · 2024-11-20
>
> > **Diversity of Kaggle Tasks.**
> - Thank you for raising this concern. While it is true that many Kaggle competitions focus on domains like retail and finance, the platform also hosts challenges from a wide range of other fields, as reflected in the dataset we collected. For instance: 1) Healthcare and Life Sciences: Competitions like Ventilator Pressure Prediction and COVID-19 Global Forecasting provide tasks directly related to healthcare and pandemic modeling. 2) Education and Natural Language Processing: Examples include CommonLit Readability Prize and Learning Agency Lab Automated Essay Scoring, which focus on NLP and educational challenges. 3) Environment and Urban Planning: Tasks like See Click Predict Fix address urban planning and environmental issues. 4) Gaming and Simulations: The Conway's Reverse Game of Life challenge explores complex systems and algorithms. 5) Miscellaneous Domains: Competitions such as TMDB Box Office Prediction and Spaceship Titanic cover unique topics, including entertainment and hypothetical scenarios. This indicates that our dataset include a diverse domains. The diverse kaggle competition metrics shown in Figure 3 of our paper also show the diversity of our tasks
> - Besides, the user can further collect because our dataset can be continually updatable for the newly released competition and othter data competition platforms, which can further enhance the diversity of the Kaggle data.
> - We will revise the manuscript to emphasize the breadth of domains represented in our dataset.
>
> > **Explain why RPG can reflect actual performance across tasks.**
> - Thank you for your comment. The Relative Performance Gap (RPG) metric is designed to provide a **normalized measure of model performance across various data types and task complexities**, allowing for a consistent evaluation framework within DSBench. Unlike raw metrics in the kaggle tasks such as accuracy and RUC, RPG accounts for performance differences by comparing the model's results against both a baseline and the best-known human expert performance for each task. This approach enables RPG to highlight how closely a model approaches expert-level proficiency, regardless of the specific data domain or task difficulty.
> - RPG is particularly valuable in DSBench's diverse setting, where tasks vary widely in complexity and metric requirements (e.g., ROC AUC for classification tasks versus Root Mean Squared Error for regression tasks). By normalizing performance relative to the best achievable score and adjusting for task-specific baselines, RPG captures nuanced differences in model efficacy. This makes RPG a robust metric for assessing agents’ ability to generalize across the wide range of challenges that data science tasks present, from data analysis to end-to-end modeling.
>
>
> [1] MLAgentBench: Evaluating Language Agents on Machine Learning Experimentation.

---

> > ### Author Response · Authors · 2024-11-21
> >
> > Thank you for your valuable feedback. As the ICLR public discussion phase will be ending in a few days, I wanted to check if my previous response has addressed your concerns. If it has, I would greatly appreciate it if you could consider updating your score accordingly. Thank you again for your time and insights!

---

### Comment · Area_Chair_6jR1 · 2024-11-25

Dear reviewers,

As the deadline for discussion is ending soon. Please respond to the authors to indicate you have read their rebuttal. If you have more questions, now is the time to ask.

AC

---

### Meta-Review · Area_Chair_6jR1 · 2024-12-16

**Metareview:**

The paper introduces DSBench, a comprehensive benchmark for evaluating data science agents through 540 tasks sourced from ModelOff and Kaggle competitions, offering a significant contribution to the field of AI-driven data science evaluation (Reviewers TVnT and EZxL). The benchmark's key strengths include its realistic task environment, covering 466 data analysis and 74 data modeling tasks, and the novel Relative Performance Gap (RPG) metric (Reviewer cMgg). Reviewers consistently highlighted the benchmark's importance in revealing substantial performance gaps between current AI models and human data scientists, with the best agent achieving only 34.12% accuracy in data analysis tasks (Reviewer TVnT). The benchmark's originality lies in its comprehensive approach, integrating multi-table reasoning, long-context understanding, and multimodal task backgrounds, making it a pioneering effort in systematically evaluating AI's data science capabilities (Reviewer TVnT).

Despite the overall positive reception, several weaknesses were identified. Reviewers raised concerns about the benchmark's limited data diversity, noting that the tasks are primarily finance-focused (Reviewer XL8J), and questioned the robustness of the evaluation methodology (Reviewer EZxL). The potential for data leakage through model pre-training was a significant concern, with one reviewer highlighting the risk of benchmark tasks being inadvertently incorporated into future model training data (Reviewer EZxL). Additionally, reviewers suggested improvements in statistical rigor, such as reporting confidence intervals, conducting bootstrap analysis, and providing more detailed explanation of the evaluation process (Reviewer TVnT). The paper also received critiques about its presentation, with some reviewers noting difficulties in understanding the taxonomy and requesting clearer explanations of data categorization and evaluation metrics (Reviewer XL8J). Despite these limitations, the majority of reviewers rated the paper as marginally above or at the acceptance threshold, recognizing its potential to drive advancements in automated data science research.

**Additional Comments On Reviewer Discussion:**

Authors have sufficiently addressed the comments in a point-by-point manner. All reviewers indicated that they were satisfied.

---

### Decision · Program_Chairs · 2025-01-22

Accept (Poster)